# Impaired Plakophilin-2 in obesity breaks cell cycle dynamics to breed adipocyte senescence

Aina Lluch [1,2], Jessica Latorre[1,2], Angela Serena-Maione [3], Isabel Espadas [4], Estefanía Caballano-Infantes [1,2], José M. Moreno-Navarrete [1,2], Núria Oliveras-Cañellas [1,2], Wifredo Ricart[1,2], María M. Malagón [2,5], Alejandro Martin-Montalvo [4], Walter Birchmeier [6], Witold Szymanski [7], Johannes Graumann [7], María Gómez-Serrano [8], Elena Sommariva [3], José M. Fernández-Real [1,2,9,10] & Francisco J. Ortega [1,2,10] ✉

Plakophilin-2 (PKP2) is a key component of desmosomes, which, when defective, is known to promote the fibro-fatty infiltration of heart muscle. Less attention has been given to its role in adipose tissue. We report here that levels of PKP2 steadily increase during fat cell differentiation, and are compromised if adipocytes are exposed to a pro-inflammatory milieu. Accordingly, expression of PKP2 in subcutaneous adipose tissue diminishes in patients with obesity, and normalizes upon mild-to-intense weight loss. We further show defective PKP2 in adipocytes to break cell cycle dynamics and yield premature senescence, a key rheostat for stress-induced adipose tissue dysfunction. Conversely, restoring PKP2 in inflamed adipocytes rewires E2F signaling towards the re-activation of cell cycle and decreased senescence. Our findings connect the expression of PKP2 in fat cells to the physiopathology of obesity, as well as uncover a previously unknown defect in cell cycle and adipocyte senescence due to impaired PKP2.

Plakophilin-2 (PKP2) is a component of desmosomes that participates in the intercellular coupling, linking cadherins to the cytoskeleton[1], and maintaining cell–cell adhesion and tissue cohesion[2]. PKP2 also interacts with voltage-gated sodium channel complexes[3], which may explain impaired sodium current in subjects carrying missense mutations in the *PKP2* gene[4]. In addition to its role as a structural protein and the electrophysiological consequences of defective PKP2 in cardiac cells[5], this member of the armadillo-repeat protein family is a component of the connexome that serves as a key scaffold in intracellular signaling, including the control of transcriptional processes in cardiac cells[6], and the regulation of β-catenin, a major participant in Wnt signaling[7]. Alterations of the latter under defective PKP2 leads to a chain of events that may disrupt the myocyte transcription program, fostering the fibro-fatty infiltration of heart muscle[8,9]. Concomitantly,

[1]Department of Diabetes, Endocrinology and Nutrition, Institut d'Investigació Biomèdica de Girona (IDIBGI), Girona, Spain. [2]CIBER de la Fisiopatología de la Obesidad y la Nutrición (CIBEROBN), Instituto de Salud Carlos III (ISCIII), Madrid, Spain. [3]Unit of Vascular Biology and Regenerative Medicine, Centro Cardiologico Monzino IRCCS, Milan, Italy. [4]Centro Andaluz de Biología Molecular y Medicina Regenerativa (CABIMER), Consejo Superior de Investigaciones Científicas (CSIC), University Pablo de Olavide, Seville, Spain. [5]Department of Cell Biology, Physiology and Immunology, Instituto Maimonides de Investigación Biomédica de Cordoba (IMIBIC), University of Cordoba, Reina Sofia University Hospital, Cordoba, Spain. [6]Max Delbrück Center for Molecular Medicine, Berlin-Buch, Germany. [7]Institute of Translational Proteomics, Biochemical/Pharmacological Centre, Philipps University, Marburg, Germany. [8]Institute for Tumor Immunology, Center for Tumor Biology and Immunology, Philipps University, Marburg, Germany. [9]Department of Medical Sciences, School of Medicine, University of Girona, Girona, Spain. [10]These authors jointly supervised this work: José M. Fernández-Real, Francisco J. Ortega. ✉e-mail: fortega@idibgi.org

PKP2-dependent variations in the transcriptome of cardiac cells have been studied in the context of exaggerated lipogenesis, metabolic derangement, and the acquisition of adipocyte traits[10]. Lately, the deleterious consequences of defective PKP2 in the myocardium have been recognized to also include the disruption of intracellular calcium homeostasis, and a higher propensity to arrhythmias in a cardiomyocyte-specific chemically-induced *Pkp2* knockout mouse model[11], matching the fibro-adipose differentiation of *PKP2*-deficient cardiac stromal cells[12]. There, the lack of PKP2 in hearth also compromised the transcription of genes that are required to maintain insulin signaling, underscoring the importance of PKP2 not only as a scaffolding protein but also as directly involved in metabolic processes[10]. Overall, these plethora of findings supports the *PKP2* gene-related origin of at least one form of the arrhythmogenic right ventricular cardiomyopathy[13,14], a heart disease characterized by progressive substitution of cardiomyocytes by adipocytes and fibrotic tissue[15]. Yet, fundamental knowledge regarding the role of PKP2 in other cell types and metabolically relevant tissues has, up to now, been neglected. Here, we present results supported by an exhaustive analysis of primary human adipocyte cultures, genome-wide transcriptomic, and proteomics, as well as cross-validations in transversal/longitudinal human patient cohorts to report evidence of a previously unknown biological function of the armadillo-repeat protein PKP2. We show that PKP2 is canonically expressed in human adipose tissue, specifically in adipocytes, and that diminished PKP2 in subcutaneous fat is an exclusive feature of obesity, returning to the levels found in lean subjects upon mild-to-intense weight loss. Then, we manipulated PKP2 levels in differentiated adipocytes to gain a better understanding of the impact of decreased PKP2 in patients with obesity, delineating a critical role in fat cell maintenance and adipocyte commitment. Collectively, our data show expression patterns of PKP2 in adipose tissue mostly related to obesity and adipocyte function, and indicate that this particular component of the desmosome not only represents an important regulator of adipogenic stimuli, but also that the partial loss of PKP2 acts as a critical driver of a defective cell cycle in adipocytes, and the resulting deregulated adipocyte senescence in subjects with obesity.

## Results

### Unexpected upregulation of PKP2 during adipogenesis

As defective/decreased Plakophilin-2 (PKP2) in myocytes results in a shift towards an adipocyte phenotype[16,17], our hypothesis anticipated the downregulation of PKP2 acquiescent for the canonical differentiation of fat precursor cells into lipid-containing mature adipocytes. Thus, we examined dynamic adaptations affecting PKP2 during bona fide adipogenesis. To characterize the changes that occur in PKP2 during in vitro differentiation of fat precursor cells (Fig. 1a), we compared measures of gene expression in subcutaneous (SC) and omental (OM) preadipocytes obtained from a female subject with a body mass index (BMI) of 26.3 kg/m², as described in ref. 18. SC preadipocytes from a female subject with a healthy weight (BMI <25 kg/m²) and a female subject with obesity (BMI >30 kg/m²) at a comparable age (35–40 years) were also cultured and stimulated with adipogenic conditions. These assays resulted in two distinct patterns for *PKP2* gene expression, segregated by fat depot and the obesity state, and matching those of adipogenesis-regulated genes, such as *ADIPOQ*, *PLIN1*, and *FASN* (Fig. 1b, c). Unexpectedly, steadily increasing *PKP2* was found in differentiating OM and SC adipocytes, with expression levels reaching an apex in terminally differentiated adipocytes (day 14). The effect was more pronounced in SC adipocytes, which surpassed the higher expression levels found in OM adipocyte precursors at early stages (days 0 and 2) of in vitro differentiation (Fig. 1b). Also, the monitoring of *PKP2* in SC preadipocytes from female donors with or without obesity showed a higher expression in differentiating "lean" adipocytes, as compared to fat cell progenitors from the female donor

with obesity (Fig. 1c). To provide more context on the regulation of *PKP2* in fat cells, we revisited global deep-sequencing analyses and gene expression profiles obtained during white adipocyte differentiation in conventional 2D cultures[19], as well as results from a 3D cell culture platform[20]. These analyses consistently yielded increased *PKP2* (but not *PKP1*, *3* or *4*) during adipogenesis (Fig. S1a, b), as evident from increased steady-state messenger (m)RNA counts in partially (7th day) and fully (14th-day post-induction) differentiated human adipocytes (Fig. S1c). As differential transcriptomes assessed in cardiomyocytes have linked defective PKP2 to inflammation[21], we proceeded to challenge SC adipocytes with 2% macrophage media (MM) and macrophage lipopolysaccharide (LPS)-conditioned media (MCM), mimicking the inflammatory environment in the obese adipose tissue (Fig. 1d), as explained in reference [22]. Notably, the impact of macrophage-derived cytokines compromised the expression of *PKP2* in our primary adipocyte cell cultures (Fig. 1e), which agrees with the changes observed in the fat cell strain Simpson-Golabi-Behmel syndrome (SGBS)[23] (Fig. S1d). In the embryonic fibroblast 3T3-L1 mouse cell line, the hormonal transition to an adipocyte phenotype was as well characterized by continuous upregulation of PKP2 protein (Fig. 1f, g) and mRNA (Fig. 1h) levels, in parallel to the significant increase in lipid content during adipogenesis. As a matter of fact, and in support of our observations, *PKP2* was the sole member of the plakophilin gene-coding family systematically captured in the nucleus of murine (Fig. S1e) and human (Fig. S1f) adipocyte subclusters, according to the deconvolution analysis performed by Emont and co-workers in their single-cell atlas of human and mouse white adipose tissues[24]. Overall, these results suggest that, contrary to its role in cardiac myocytes, expression of *PKP2* does not oppose the differentiation of bona fide adipose cell precursors, but instead represents a hallmark of the adipocyte phenotype.

### Impaired PKP2 in adipocyte-precursor cells boosts adipogenesis

Previous lines of evidence have linked the transient disruption of PKP2 to metabolic disturbances and impaired insulin signaling[11], a pivotal mechanism in regulating energy storage and fat cell differentiation. To elucidate whether Pkp2 in 3T3-L1 is required for adipocyte differentiation, we investigated the consequences of PKP2 deficiency in 3T3-L1 preadipocytes in the context of the differentiation into lipid-containing fat cells (Fig. 1f). Initially, we observed impaired growth prior to confluence (Fig. 1i), recapitulating multiple lines of evidence linking PKP2 to cell proliferation[25–27]. When, however, enforcing an adipocyte phenotype in confluent cultures of engineered adipocyte progenitors, decreased *Pkp2* resulted in higher amounts of lipids (as estimated by Oil Red staining) (Fig. 1j, k), and elevated expression of adipocyte biomarkers (e.g., *Adipoq*, *Glut4*, *Plin1*) in both unstimulated preadipocytes (Fig. 1l) and differentiated adipocytes (Fig. 1m). The effect was notably also observed in primary human SC adipocytes subjected to lentiviral-mediated *PKP2* reduction during adipogenesis (Fig. S1g). In contrast, expression of interleukin 6 (*Il6*) declined in response to the knockdown of *PKP2* in differentiating mouse and human adipocytes (Fig. 1l, m and S1g, respectively). This is particularly intriguing as the loss of PKP2 has been shown to elevate p38 mitogen-activated protein kinase (MAPK) signaling, which potentially exacerbates the effects of fibrosis and inflammation in cardiomyocytes[28], while also representing a potent inhibitor of adipocyte development[29]. In adipocyte progenitors, this appears not to be the case, as defective *Pkp2* in preadipocytes did not exhibit elevated but lowered p38 MAPK signaling, and higher levels of phosphorylated Akt and Fatty acid synthase (FAS) while undergoing adipogenesis (Fig. S1h, i), thus enhancing the adipogenic process[30]. Altogether, collected evidence in fat precursor cells indicates that the expression of *PKP2* rises in parallel with the formation of new adipocytes, and that reducing its levels enhances the adipogenic transformation of adipocyte progenitors as it does in cardiac mesenchymal stromal cells[17].

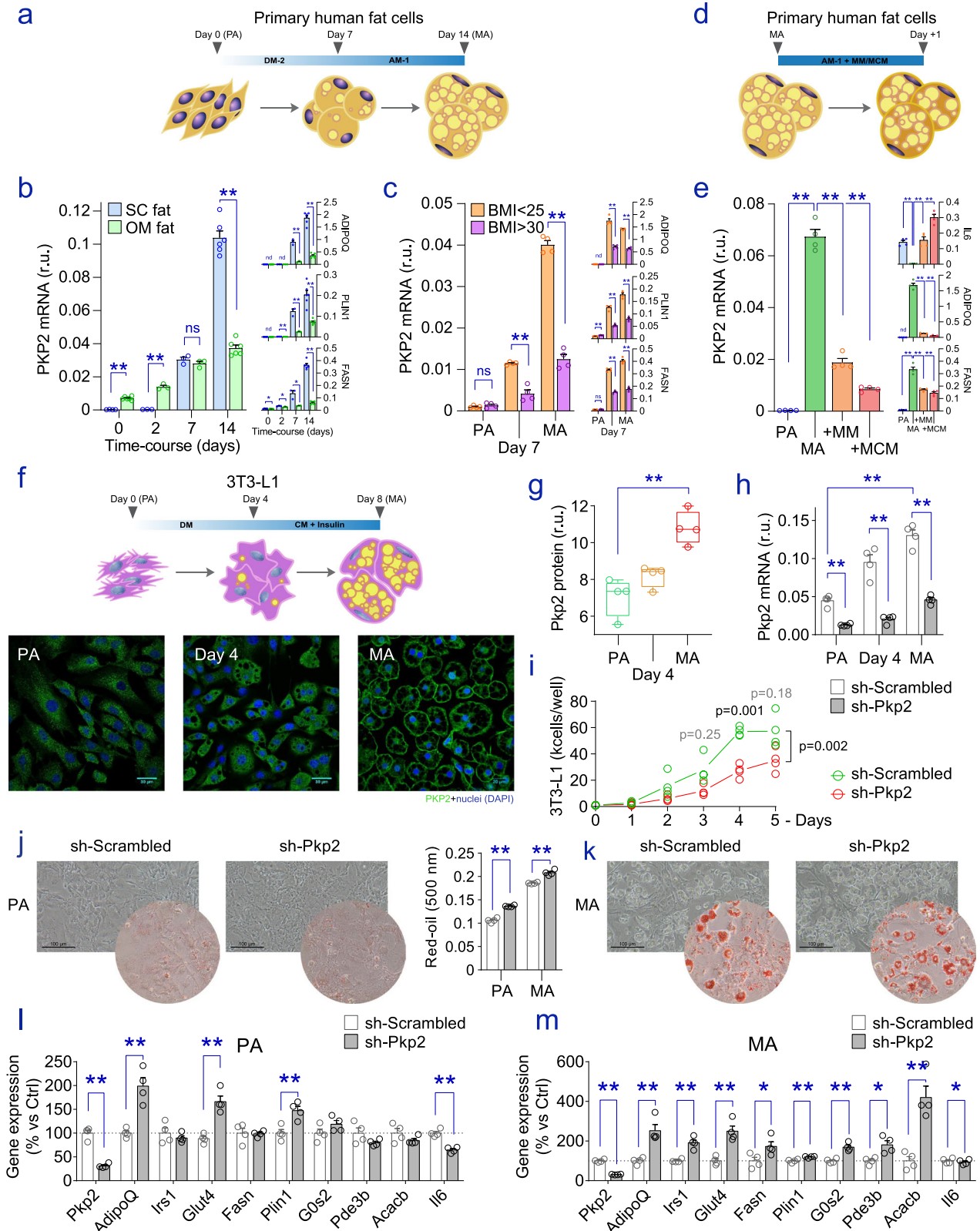

## Age and obesity exert depot-specific effects on adipose PKP2

Next, we sought to study the expression patterns of *PKP2* in human SC and OM adipose tissues. Notably, during the analysis of adaptive variation affecting the transcriptome of SC adipose tissue[31], we observed that *PKP2* mRNA was strongly increased upon weight loss (Fig. S2a). Indeed, its potential value as a biomarker of adipose function was corroborated by the fact that, amidst all plakophilins, only *PKP2*

recovered to expression levels found in age-matched non-obese participants, in parallel with the loss of significant amounts of fat mass lasting 2 or 5 years after bariatric surgery[32] (Fig. S2b). To confirm these changes, we also checked the results from two independent next-generation sequencing (NGS) analyses. The Diet, Obesity and Genes (DiOGenes) study[33] assessed 382 transcriptomes in the SC adipose tissue of participants under a low-calorie diet (800 kcal/day) for 8

**Fig. 1 | PKP2 in putative fat cells. a** Pipeline diagram of in vitro cultured human PA growing into differentiated lipid-containing MA. DM-2 (first) and AM-1 (second week) stand for differentiation and adipocyte media, respectively (see in Methods). Expression of *PKP2* and adipogenic biomarkers (*ADIPOQ, PLIN1, FASN*) during differentiation of fat progenitors from (**b**) SC (*n* = 4, 3, 3, and 6) and OM (*n* = 6, 3, 3, and 6 biological replicates for days 0, 2, 7, and 14 of hormonal stimuli, respectively) adipose tissues of the same donor, and from **c** the SC adipose tissue of one female donor without (BMI < 25 kg/m²) and another one with obesity (BMI >30 kg/m²) (*n* = 4/timepoint/group). **d** Impact of macrophage (MM) and macrophage LPS-conditioned media (MCM) on (**e**) the expression of *PKP2* and inflammatory (*IL6*) and adipogenic (*ADIPOQ, FASN*) biomarkers in terminally differentiated MA (*n* = 4/ group). **f** The diagram displays the conditioned differentiation of 3T3-L1 cells into MA, including 4 days under the influence of a differentiation medium (DM), and 4 days of basal preadipocyte culture media (CM) with insulin. Below, the immunofluorescent staining of 3T3-L1 cells exposed to the adipogenic cocktail. Scale bars show sizes of 30 (PA and Day 4) or 20 (MA) μm. **g** Levels of PKP2 protein in 3T3-L1

cells while developing into MA, as assessed by western blot. The box plot shows the center line at the median, upper and lower lines bound at 75th and 25th percentiles, respectively, and whiskers at minimum and maximum values. **h** Expression of *Pkp2* in differentiating 3T3-L1 cells infected with a retrovirus carrying an empty vector (sh-Scrambled) or sh-*Pkp2*. **i** The resulting engineered 3T3-L1 cells in growth conditions showed apparent variations regarding their proliferation rate. **j, k** Increased amounts of lipids and **l, m** gene expression patterns were suggestive of enhanced adipogenesis in sh-*Pkp2* cells maintained under either **j, l** non-adipogenic (PA) and **k, m** differentiating conditions (MA). The scale bar in optical microscope images denotes 100 μm length. Plots for 3T3-L1 cells show data assessed in *n* = 4 biological replicates/conditions. In all bar plots, results are presented as mean ± SEM. Statistical significance was assessed by two-tailed Fisher's exact *t*-test, and by adjusted ANOVA after applying Šidák's correction to repeated measures (cell growth). r.u. relative units, nd non-detectable, ns non-significant; *$p < 0.05$; **$p < 0.01$. Source data are provided as a Source data file.

months (65% women; weight loss of 11.9 ± 7.1%). The Kuopio OBesity Surgery (KOBS) is a prospective observational study of the metabolic consequences of bariatric surgery[34]. There, 172 eligible individuals with obesity underwent gastric bypass, and SC bulk sequencing data were obtained at the baseline and 12 months of follow-up (72% women; weight loss of 22.4 ± 7.7%). In line with our observations in refs. [31,32], these analyses confirmed increased SC *PKP2* after the loss of significant amounts of fat (Fig. 2a), with its expression being, on average, more than 3 points higher following weight loss (16.9 ± 13.9 and 13.4 ± 11.1 c.p.m., respectively), as depicted by results of the DiOGenes study (Fig. S2c). Based on this data, we used real-time PCR and TaqMan hydrolysis assays to analyse the expression of *PKP2* in two independent cohorts. Cohort 1 included 24 lean and 20 age-matched women with morbid obesity followed for an average of approximately 2 years after bariatric surgery, which resulted in pronounced loss of fat mass and metabolic improvement (Table S1). Cohort 2 consisted of 219 individuals (78% women) with a wide range of weight and glucose tolerance (Table S2). In both cohorts, SC abdominal biopsies of adipose tissue were obtained, in volunteers with obesity of cohort 1 both at baseline and follow-up. Paired (*n* = 90) and unpaired OM fat samples were also studied in cohort 2. In confirmation of the microarrays and NGS results, expression of SC *PKP2* was increased during major weight loss (100% women; weight loss of 27 ± 9.2%) by a radical reduction in calorie intake after bariatric surgery, reaching expression levels equivalent to those observed in women with healthy body weight (Fig. 2b and Table S1). Increased amounts of SC PKP2 were evident as well at the protein level in obese patients upon weight loss (Fig. 2c). In agreement with our longitudinal findings, *PKP2* was found to be diminished in SC adipose compartments of patients with obesity, especially in subjects with attendant impaired fasting glucose (Fig. 2d and Table S2), and thus inversely associated with BMI, the expression of leptin (*LEP*) (Fig. 2e), and percent fat mass, as well as biomarkers of an impaired metabolic control (Table S3). In contrast to results in SC adipose tissue, measures of *PKP2* in OM fat revealed no relationship with BMI (Fig. 2f, g), yet slightly decreased expressions in non-obese subjects with IGT (Fig. 2f), and the inverse association with OM *LEP* (Fig. 2g), were confirmed. In fact, the expression of *PKP2* in OM adipose tissue was independent of the expression levels obtained in paired SC fat samples, while inversely associated with age, LDL cholesterol, and glycated hemoglobin as surrogates of age-related impaired glucose and lipid metabolism (Table S3). On the other hand, fluorescent antibodies and confocal microscopy applied to slides of SC adipose tissue revealed PKP2 immunolabeling primarily located in adipocyte membranes, while being more prevalent in the interstitial compartment of OM fat (Fig. 2h). Consistently, measures of *PKP2* gene expression taken in ex vivo isolated mature adipocytes (MA) and the stromal vascular cell (SVC) fraction of morbid obese patients were suggestive of different cell populations growing within OM (but not SC) SVC, as shown by our

RT-PCR results (Fig. 2i) and the single-cell RNA sequencing of ref. 35 (Fig. S2d). This matches the enrichment of *PKP2* mRNA in SC adipocytes, while being more likely associated with the expression of biomarkers of mesothelial cells within OM fat (Fig. S2e)[36]. However, because these observations in bulk adipose tissue and adipose-derived cell samples were not population-based, we cannot exclude that the apparent relationship with obesity (results in SC) and other clinical characteristics (OM and SC) relies on the abundance of specific fat cell subgroups regulated during obesity. Nevertheless, our experiments in vitro, together with the systematic scrutiny of multiple human datasets and observations made in patients with obesity following weight loss, support the connexion between low PKP2 in SC adipose tissue/ adipocytes and the burden of obesity/ inflammation. The exact mechanism whereby sex, age, and/or metabolic status may influence the expression of *PKP2* in the even more complex cellularity of OM adipose tissue is a key question we seek to answer in our future studies. Notwithstanding this, *PKP2* in SC (Fig. S2f) and OM (Fig. S2g) adipose tissue was inversely associated with the expression of 11β-hydroxysteroid dehydrogenase type 1 (*HSD11B1*) and secreted phosphoprotein 1 (*SPP1*, also known as osteopontin), while positively correlating with the expression of the very long-chain acyl-CoA synthetase encoded by the solute carrier family 27 (fatty acid transporter), member 2 (*SLC27A2*) gene (Table S3). In fact, multiple linear regression analyses revealed that *SLC27A2* was the best predictor of *PKP2* gene expression levels in samples of SC (53.1% of the variance, β = 0.68) and OM (26.6%, β = 0.58; both $p < 0.0001$) adipose tissue, after being corrected for gender, age, and BMI. Noteworthy, SC (but not OM) *PKP2* was directly associated with *FASN* and *CIDEA*, while inversely related to *TNF* (Fig. S2h), pointing to its relationship with the inflammatory and metabolic state of fat cells. Conversely, *PKP2* in OM (but not in SC) adipose tissue was associated with the expression levels of depot-specific[37] thermogenic/beige-related genes such as *NRG4*[38,39] and *TMEM26*[40] (Fig. S2i), which was in agreement with the enrichment revealed previously in the SVC fraction of OM fat samples[38], and mostly related to the ontogenetic presence of mesothelial cells within this fat depot[41]. Altogether, our compiled clinical and biochemical data revealed different expression patterns affecting *PKP2* in human fat depots as a surrogate of obesity and impaired metabolism, adipose cell populations (in OM), and adipocyte commitment (in SC adipose tissue).

## Impaired PKP2 in human adipocytes deeply alters transcriptomes

As the changes occurring in inflamed adipocytes were consistent with a significant loss of SC *PKP2* in patients with obesity, we next transitioned our findings to complementary in vitro assays. To survey the functional contribution of reduced *PKP2* to canonical fat cell commitment, we challenged human adipocytes with a synthetic silencing

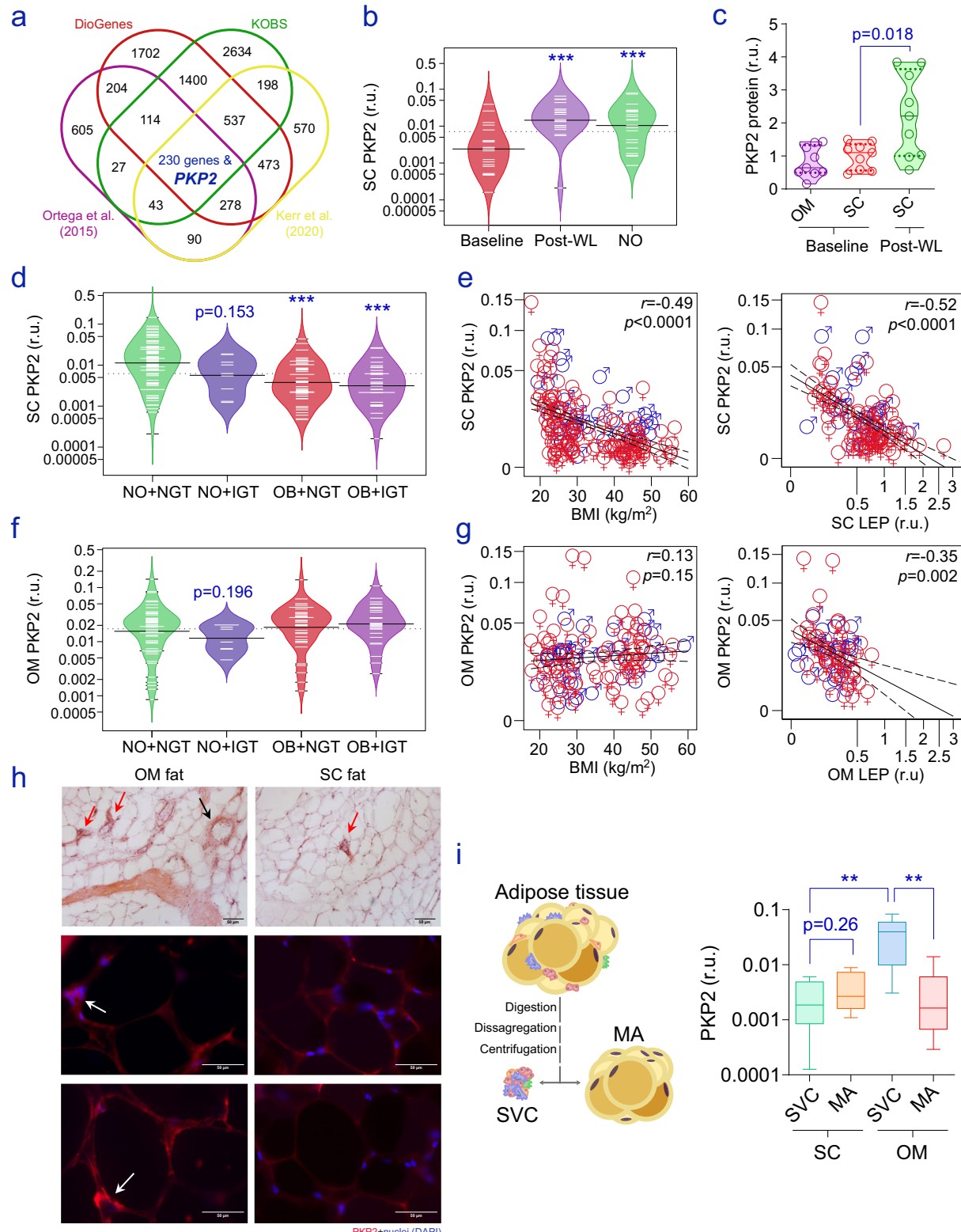

(si)RNA directed against *PKP2* (si-*PKP2*). As the dynamic turnover of desmosomal proteins in cell cultures is mostly unknown[42], we also used lentiviral-mediated transfection techniques in primary human adipocytes with viruses expressing either silencing short hairpin (sh) RNAs for *PKP2* (sh-*PKP2*) or non-silencing (NS) control (Fig. 3a). Using this protocol, a more powerful, robust, and long-lasting modification of targeted genes may be achieved in adipocytes, as compared to siRNAs[43]. Together, these systems allowed us to control the timing of

impaired PKP2 levels in independent cultures of human adipocytes (Fig. 3b). In order to investigate the landscape of genes regulated by the loss of PKP2, we performed microarray analysis on (i) lipid-containing white adipocytes transiently treated with synthetic siRNAs and non-targeting controls (NTC) for 72 h, and (ii) engineered primary human adipocytes with *PKP2* expression levels decreased for the duration of 6 days (Fig. 3c). Hierarchical clustering (Fig. 3d) and principal component analysis (Fig. 3e) revealed a high level of

**Fig. 2 | PKP2 in human adipose tissue. a** The Venn diagram shows the number of genes significantly (FDR <0.05) increased (fold-change >1.2) in four independent datasets containing transcriptional profiling of SC adipose tissue upon surgery[31,32,34] or diet-induced[33] weight loss (WL). **b** The bean plot shows significant RT-PCR variations affecting SC *PKP2* at the baseline (red) and upon surgery-induced WL (purple), and also when patients with obesity were compared to age-matched non-obese (NO) controls (see in Table S1). **c** Changes in SC PKP2 after surgery-induced WL were confirmed at the protein level (western blot) in a subset of nine participants with morbid obesity, while no significant differences were identified in paired SC-OM fat samples at the baseline. **d** SC and **f** OM *PKP2* gene expression in subjects segregated according to their BMI and glucose tolerance as NO (BMI <30 kg/m²) and participants with obesity (OB) and normal (NGT) or impaired (IGT) glucose tolerance (see also in Table S2). Association between **e** SC and **g** OM *PKP2*, BMI, and the expression of leptin (*LEP*) in each depot (see these and other associations in

Table S3). Statistical significance for differences between groups of subjects was determined by one-way ANOVA followed by Tukey's honestly significant difference (HSD) post hoc tests (cross-sectional sample), and by Student's paired *t*-test (longitudinal studies). Spearman's rank-order correlation tests were employed in correlation analyses. **h** Haematoxylin and eosin, and the immunofluorescent staining of PKP2 in SC and OM fat samples from the same donors depicted differences related to the distribution of adipocyte-specific PKP2 (SC adipose tissue), and its signal located in crown-like structures (red and white arrows) and/or blood vessels (black arrow) within OM fat. The scale bar in optical and immunofluorescent pictures denotes 50 µm length. **i** Expression of *PKP2* in ex vivo isolated SVC and MA from SC and OM fat samples (*n* = 9 and 18, respectively). The box plot shows 75th to 25th percentiles with the median, and whiskers at maximum and minimum values. Statistical significance was assessed by a two-tailed Fisher's exact *t*-test. r.u. relative units; *$p < 0.05$; **$p < 0.01$. Source data are provided as a Source data file.

homogeneity in each group, and clear separation between adipocytes with impaired *PKP2* and reference controls. Subsequently, a total of 20,893 transcripts contained in the Affymetrix oligonucleotide microarray were tested for differential expressions in each experiment. Stringent threshold criteria (i.e., adjusted false discovery rate (FDR) *p* value <0.05 plus fold-change >[1.2]) resulted in 463 differentially expressed transcripts (66 up and 397 downregulated) between si-*PKP2* and NTC human adipocytes, while 1811 transcripts (890 up and 921 downregulated) highlighted the stronger impact of the 6 days-lasting lentiviral-mediated knockdown of *PKP2*. Gene Set Enrichment Analysis (GSEA)[44] (Fig. 3f, g), Metascape[45] (Fig. S3a, b), and Ingenuity Pathway Analysis (IPA) (Fig. S3c) provided consistent evidence that the genes enriched in these models of loss-of-function were mostly related to proliferation and cell-cycle control (i.e., G2M checkpoint and mitotic spindle), and/or were genes coding for molecules downstream the activity of E2F transcription factors (E2F targets) (Fig. S3d, e). Notably, these hallmarks were compromised in both assays, with overlaps of 45% (49 coincidences amidst the leading-edge subset within the G2M checkpoint gene set) and 50% (52 genes shared within the E2F target genes), respectively (Fig. S3f), which summarizes the downregulation of 25 of the genes that contribute most to these hallmarks (Fig. S3g). Finally, heat maps showed the consistency of significant changes affecting a common set of 68 transcripts modified in both assays (Fig. 3h, i). Intriguingly, the increased expression of four unique genes was observed in each experimental setting. These were genes coding for the TGFβ1-activating Integrin β8 (*ITGB8*)[46], Inositol monophosphatase 2 (*IMPA2*), DNA damage-inducible transcript 4 (*DDIT4*, also known as *REDD1*), and Pappalysin 1 (*PAPPA*), labeled along with *PKP2* in Fig. 3c. It is worth noting that upregulated IMPA2 may inhibit the phosphorylation of Akt/mTORC1, thus compromising proliferation and cell survival[47]. On the other hand, chronic expression of the ubiquitous protein Regulated in development and DNA damage response 1 (or REDD1) may transition the pathogenesis of several diseases[48], being also reported its ability to modulate energy balance and metabolism[49]. Finally, the inhibition of the Pregnancy-associated plasma protein-A (or Pappalysin 1) has been proposed as a therapeutic strategy to prevent some of the metabolic sequelae of obesity[50], and even to promote healthy longevity[51]. Thus, discontinued *PKP2* in obese/inflamed adipocytes may contribute to the loss of fat cell plasticity, deranged adipose tissue function, and the deleterious consequences of obesity.

## Impact of defective PKP2 in fat cells disclosed by untargeted proteomics

As many of the functions ascribable to PKP2 are dependent on a sequence of structural changes at the protein level, we conducted quantitative proteomics to further substantiate our transcriptomic observations following the knockdown of *PKP2* (Fig. 4a). First, we determined the proteomes of control and si-*PKP2*-targeted human adipocytes and preadipocytes using high-resolution liquid

chromatography-tandem mass spectrometry. Initially, a data-independent acquisition strategy and analysis using DIA-NN[52] identified 8214 protein groups. After dropping 52 peptides without replication within subgroups, and upon filtering against 1684 proteins with less than two peptides, 6478 protein groups were retained for further analysis (Fig. S4a, b). Hierarchical clustering (Fig. S4c) and the correlation plot (Fig. S4d) of results obtained in each proteome yielded clear segregation of biological replicates into their respective subgroups. Principal component analysis (PCA) also showed that adipocytes (Fig. 4b) and preadipocytes (Fig. 4c) challenged with si-RNAs targeting *PKP2* were readily distinguished from corresponding controls. Then, differentially abundant proteins (DAPs) were assessed by employing Bayesian moderated *t*-test (*p* value <0.05). Amongst these, 461 DAPs (346 up and 115 downregulated) in adipocytes (Fig. 4d), and 406 (219 up and 187 downregulated) in preadipocytes (Fig. 4e) were shortlisted after correcting for multiple hypothesis testing (FDR). Of note, a number of the proteins that were downregulated in adipocyte cultures with defective PKP2 were associated with the gene ontology terms *Cell cycle* (GO:0007049) and *Focal adhesion* (GO:0005925), while DAPs increased in this experimental setting pointed at the *Cell response to stress* (GO:0033554), as highlighted in Fig. 4f, the supplemental Table S4, and in Fig. S5a, S5b. Conversely, si-*PKP2* preadipocyte proteomes were characterized by the lower abundance of proteins related to the *Response to stress* (GO:0006950), also accompanying impaired *Cell cycle* (Fig. S5c) and *Focal adhesion* (Fig. 4g), while being over-represented for proteins of the *Carbohydrate derivative metabolic process* (GO:1901135), amongst others (Fig. S5d and Table S4). Notably, the cumulative frequency of changes affecting proteins involved in these processes (Fig. 4h) highlighted the degree of coordination between DAPs in common enriched gene ontology categories in mature adipocytes and undifferentiated precursor cells with impaired *PKP2* (Fig. S4e). Amongst these, the decreased presence of cell cycle-related key proteins such as E2F4, ERBB2, KI67 (*MKI67*), and TOPK (*TOP2A*), and altered abundance of enzymes mainly responsible for the early stages of the biosynthesis of lipids (SCD, ACLY), together with enhanced PAPP1 (*PAPPA*), IMPA2, and Insulin-like growth factor binding protein 3 (*IGFBP3*, also known as IBP3), and decreased Metal-lothionein 1 F (MT1F) protein levels (Fig. 4i), further validated our transcriptomic results in mature adipocytes, thus further suggesting a transitional state of *PKP2*-deficient precursor cells to acquire some adipocyte features under conditions that per se do not trigger the adipogenic transformation (Fig. S4f).

## Querying adipose cells of *Pkp2*⁺/⁻ mice and *PKP2*-dependent ACM patients

To confirm that defective PKP2 in sadipose tissue is mostly associated with gene expression patterns suggestive of impaired fat cell cycle, we took advantage of subcutaneous adipose tissues collected in a heterozygous (*Pkp2*⁺/⁻) *Pkp2* knockout mouse model[2,53] (Fig. 5a), and analysed the subcutaneous stromal cells obtained from fat biopsy

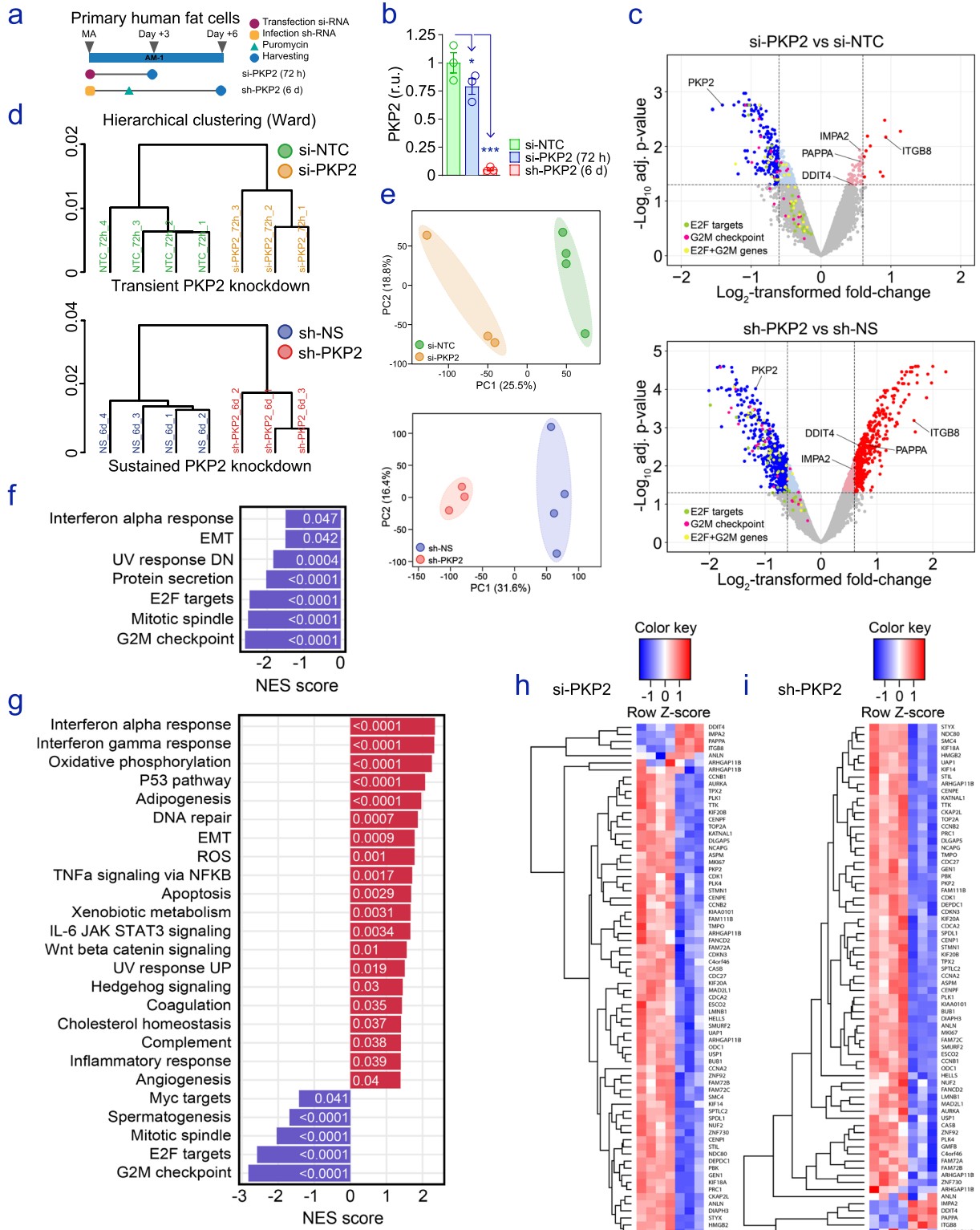

**Fig. 3 | PKP2 is an agonist of the cell cycle in human adipocytes. a** Diagram illustrating the treatments applied to fully-differentiated MA, which **b** compromised the abundance of PKP2 protein to lesser (si-PKP2) or a greater (sh-PKP2) degree (mean ± S.E.M for $n = 3$ biological replicates/group). **c** Volcano plots, **d** hierarchical clustering, **e** principal component analysis (PCA), and **f**, **g** ingenuity pathway analysis (IPA) showing and interpreting the changes found in transcriptomes of siRNA (si-PKP2) and lentiviral-mediated (sh-PKP2) *PKP2* knock-down in human adipocytes ($n = 3$ replicates/approach), when compared to their respective controls (si-NTC and sh-NS; $n = 4$). Heat maps show variations in a list of 68 common genes significantly altered in MA challenged with either **h** siRNAs or **i** the lentiviral vector-mediated construct against *PKP2*.

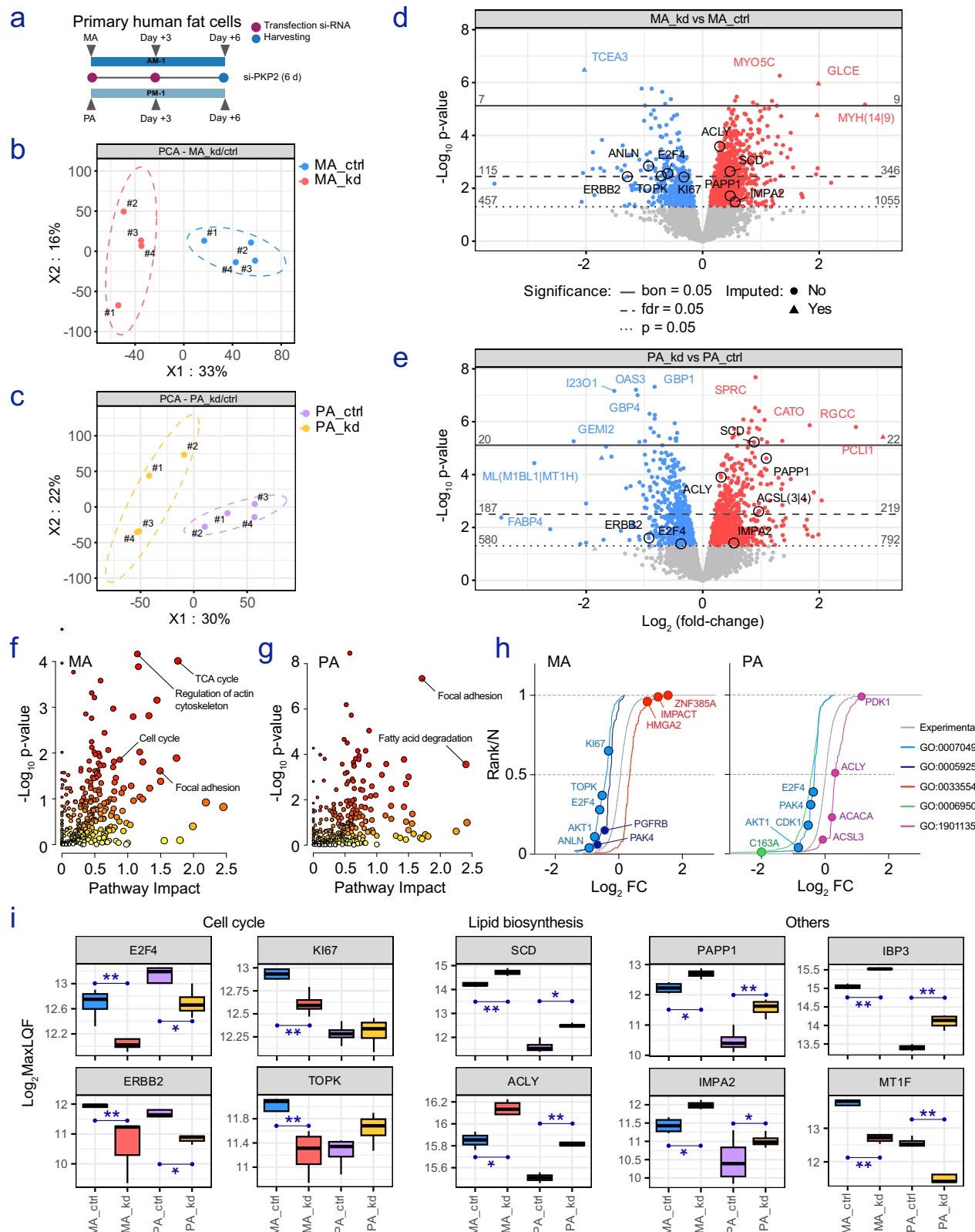

sampling in *PKP2*-dependent arrhythmogenic cardiomyopathy (ACM) patients (Table S5)[17,54], as illustrated in Fig. 5b. In *Pkp2*[+/-] mice (Table S6), the analysis of gene candidates related to cell cycle depicted a global trend towards decreasing expression levels in whole adipose tissue and ex vivo isolated adipocytes, yet no consistent effect on adipogenesis-regulated genes (e.g., *Pparg, Srebp1, Adipoq, Plin1*) were observed (Fig. 5c). In our limited human cohort, we also did not

detect significant effects on genes related to fat cell performance (*FASN, PLIN1*), senescence (*SQSTM1*), metabolic imbalance (*PAPPA, DDIT4*), or enhanced inflammation (*TNF*), but instead observed a common trend of decrease for genes belonging to cell cycle in ex vivo isolated *PKP2*-mutated adipose mesenchymal stromal cells, after correcting for age and sex (Fig. 5d). Furthermore, when these cells were plated on plastic in preadipocytes culture medium, grown until

**Fig. 4 | Proteome analysis of *PKP2*-deficient adipocytes and precursor cells.**
**a** Timeline of treatments applied to fat cell cultures in which mass spectrometry-based quantitative proteomics have been conducted. Multivariable analysis (PCA) highlighting the clustering of non-targeting control (ctrl) and knocked-down (kd) **b** mature adipocytes (MA) and **c** preadipocytes (PA). Numbers indicate biological replicates for each subgroup ($n = 4$). Volcano plots of upregulated (red) and downregulated (blue) differentially abundant proteins (DAPs) in **d** MA and **e** PA knocked down for *PKP2*. Dots in gray show proteins meeting our exclusion criteria (*p* value ≥0.05). Pathway analysis plots were created by applying MetaboAnalyst 5.0 to **f** MA and **g** PA DAPs. The *x*-axis represents the pathway impact value computed from pathway topological analysis, and the y-axis is the $-\log_{10}$ of the *P* value obtained from pathway enrichment analysis. **h** Sigmoidal plots show the cumulative frequency of $\log_2$ fold changes affecting DAPs annotated in enriched functional categories highlighted in the supplementary Table S4 and Fig. S5, including *Cell cycle* (GO:0007049) in both cell models; *Focal adhesion* (GO:0005925) and *Cell response to stress* (GO:0033554) in MA; and *Response to stress* (GO:0006950) and

*Carbohydrate derivative metabolic process* (GO:1901135) in PA. Representative DAPs of the different categories are labeled. The distribution of changes for all the proteins quantitated ($n = 6478$; gray line) is shown as a reference (Experimental). The shape of the sigmoidal curves corresponding to the enriched categories indicates the shift from the experimental distribution (left, decreased; right, increased). **i** Cell and treatment-specific differences regarding the amounts of specific proteins involved in the cell cycle, lipid biosynthesis, and factors consistently increased in the fat cell models with defective PKP2 studied in this research (Others). Relative intensity values were calculated and are expressed as $\log_2$MaxLFQ, a generic label-free quantification technology. In boxplots, the upper whisker extends from the hinge to the largest value no further than 1.5 * IQR from the hinge (where IQR is the inter-quartile range, or distance between the first and third quartiles). The lower whisker extends from the hinge to the smallest value at most 1.5 * IQR of the hinge. Statistical significance was assessed employing Bayesian moderated *t*-tests (unpaired and assuming equal variance) in comparisons of $n = 4$ biological replicates/group. *$p < 0.05$; **$p < 0.01$.

confluence, and stimulated or not with adipogenic conditions (mature adipocytes and preadipocytes), altered expression of *CDK1*, *MKI67*, *PLK4*, and *TOP2A* was verified in *PKP2*-mutated undifferentiated and differentiated adipocytes (Fig. 5e, f). *PAPPA*, *IMPA2*, and *MT1F* also showed expression patterns mostly linked with the genotype of *PKP2* (Fig. 5e, f), regardless of potential donor-related confounders (e.g., age, sex, etc.) and the overall nature of these adipose-derived mesenchymal stromal cells when forced to differentiate into adipocytes. Altogether, although the molecular changes observed do not indicate major potential metabolic dysfunctions affecting adipose cells in ACM patients, examination of our hypothesis in alternative ex vivo isolated and in vitro cultured adipocyte-precursor cells confirms that PKP2 deficiency in adipose tissue may drive the impaired expression of genes related to cell cycle, suggesting alternative pathophysiological scenarios that may contribute to *PKP2*-dependent ACM phenotype severity and warrant further investigations.

**Decreased PKP2 breaks cell cycle dynamics to breed adipocyte senescence**

The assumption that terminal differentiation of adipocytes is accompanied by permanent growth arrest was put forward in 1973[55], but has been debated ever since[56,57]. As a matter of fact, expression of canonical cell cycle markers and S-phase-associated transcripts, including cyclins and cyclin-dependent kinases, as well as members of the replication complexes such as DNA polymerases and replication-dependent histones[58], has been identified in MA, which might also be able to acquire a senescent-like state in both aging[59], impaired energy homeostasis, and obesity[57,60]. In our transcriptional profiles, we identified the strong downregulation of many cell cycle gene markers in MA with impaired *PKP2* levels (Fig. S6a), independent of the method (i.e., si-*PKP2* or sh-*PKP2*-mediated knockdown) and treatment length (3 and 6 days). Notably, the more consistent impact was found in the sh-*PKP2*-mediated assay, as MA showing defective *PKP2* lasting 6 days expressed diminished levels of G1/S/G2/M-phase transcripts when compared to control (Fig. S6a). Extending those findings to the protein level, high-throughput proteomics revealed that 84 of the proteins affected by the knockdown of *PKP2* were functionally annotated as *Cell cycle* (GO:0007049), corresponding to 17.4% of the proteins downregulated in adipocytes. Accordingly, *PKP2* deficiency led to suppression of E2F1 signaling, resulting in the impaired presence of key cell cycle-related proteins such as E2F4, ERBB2, KI67, and TOPK, amongst others. Concomitantly, microarray results also highlighted the increased expression of senescence and quiescence-associated genes in both assays, while transcripts commonly downregulated in senescent cells were decreased to some extent (Fig. S6b). To further substantiate the loss of adipocyte-specific *PKP2* in obesity to effectively break the cell cycle and trigger premature senescence, we performed additional assays, together with tests for cell cycle dynamics and

cellular senescence in human adipocytes. To this end, we designed a comprehensive pipeline (Fig. 6a) to sequentially filter preselected biomarkers through experimental validations in which we assessed their expression under (i) defective *PKP2* and (ii) sustained MCM-induced inflammation, as a proxy of the chronic low levels of macrophage-derived pro-inflammatory cytokines commonly found in obese adipose tissues[61]. We focused on the underrepresentation of G2M checkpoint and E2F target genes, a recognized feature of senescent cells[62], the increased expression of *PAPPA*, and the assessment of senescence-related and cell cycle biomarkers, including senescence-associated β-galactosidase (SA-β-gal) staining[63] and flow cytometry (Fig. S6c, d). We also evaluated changes in oxidant production, related to the loss of nuclear envelope integrity and DNA damage in *PKP2*-defective cardiac cells[64], by assessing $H_2O_2$ in the media, and conducted immunofluorescence staining of the cell cycle driver Cyclin D1 (CCND1)[65,66] and Protein Kinase Cα (PKCα), a scaffold regulator linking signal transduction pathways and cell-cycle machinery[67] that is directly influenced by the lack of PKP2[1]. These additional tests confirmed significant variations in the expression of key genes previously identified in human adipocyte cell cultures with impaired *PKP2* (Fig. 6b), mimicking to a great extent the changes resulting from an inflammatory microenvironment, also characterized by decreased *PKP2* (Fig. 6c). As expected, adipocytes challenged with 1% MCM for 6 days, as well as the sustained knockdown of *PKP2*, showed enhanced SA-β-gal activity, as revealed by β-galactosidase-staining (Fig. 6d) and flow cytometry (Fig. 6e). We further confirmed the increased presence of PAPPA at the protein level (Fig. 6f), also accompanied by the high release of $H_2O_2$ into the culture media of *PKP2*-deficient adipocytes (Fig. 6g), and found Cyclin D1 immunostaining to be significantly compromised (Fig. 6h), while the appearance of nuclear PKCα (Figures S6e, f) was higher in adipocytes that had been challenged with synthetic oligonucleotides against *PKP2*, and in those under the influence of MCM (Fig. 6i). Altogether, our observations highlighted an apparent disruption in PKCα-dependent fat cell cycle dynamics that may lead adipocyte senescence in cells subjected to activated pro-inflammatory fates, and thus, showing hindered *PKP2* expression levels.

**Recovery of PKP2 in inflamed adipocytes restores the cell cycle to avoid senescence**

To better assess the functional relevance of defective PKP2 in fat cells, a conventional plasmid-based vector containing the ORF expression clone of human *PKP2* was employed for transfection into MCM-inflamed adipocytes (Fig. 6a). This provided PKP2 even under conditions shown above to compromise native *PKP2* gene expression. In this assay (MCM + OE_PKP2), engineered human adipocyte cells containing the plasmid displayed expression patterns opposite to those found in cells challenged with siRNAs against *PKP2* or MCM, including the

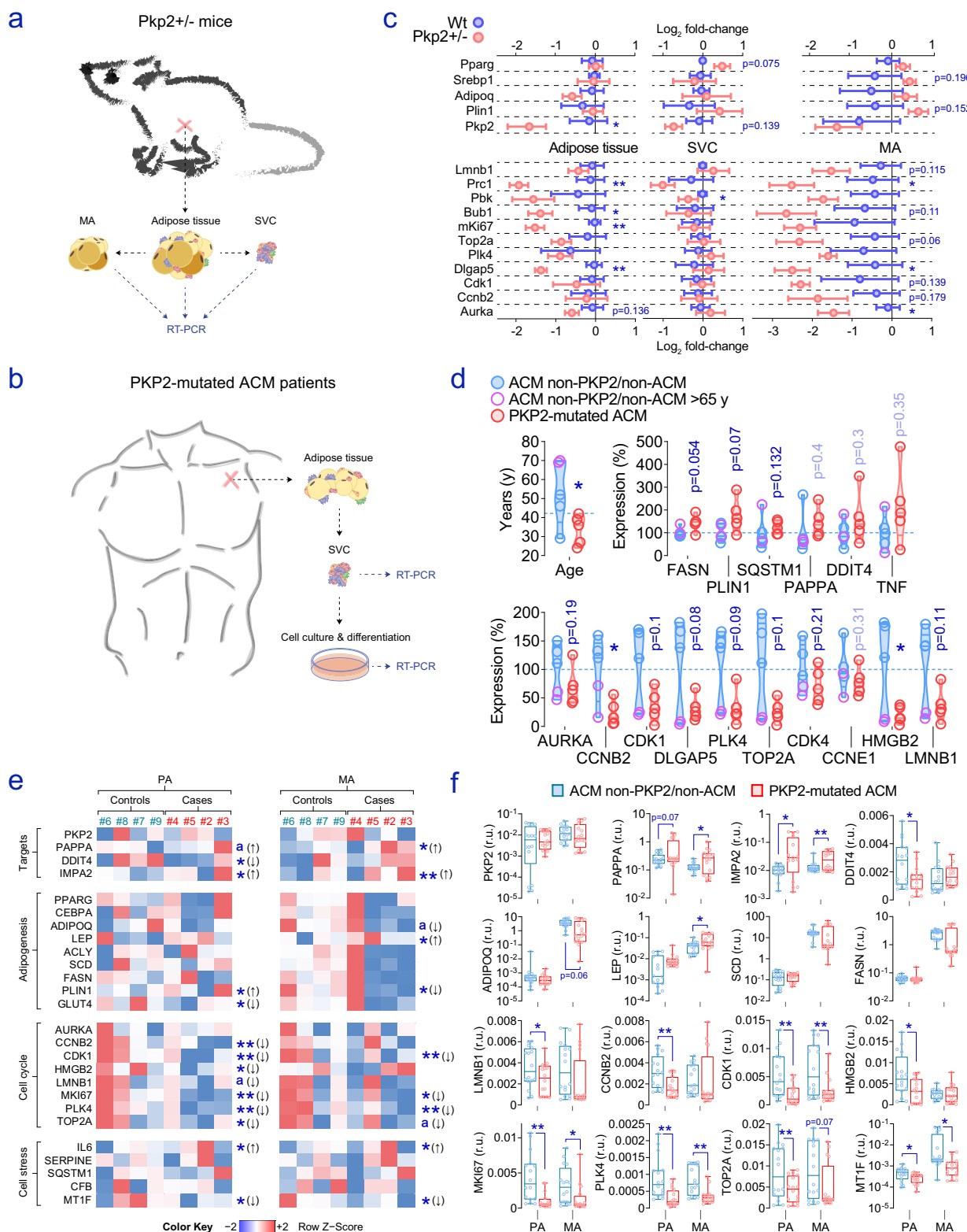

alteration of senescence-related biomarkers (e.g., decreased *ERF* and *CREG1*, and increased *HMGB2* and *LMNB1*, standing for biomarkers up and downregulated in the bulk of senescent cells, respectively), and counterbalanced the expression of cell cycle genes (e.g., *CCNB2* and *PLK4*) (Fig. 6j). Furthermore, increased SA-β-gal activity due to the inflammatory impact of effectors contained in the MCM was offset to some extent by the plasmid coding for *PKP2*, as shown by β-galactosidase-staining (Fig. 6d) and flow cytometry (Fig. 6e). $H_2O_2$

levels (Fig. 6g), impaired CCND1 (Fig. 6h, k), the appearance of nuclear PKCα (Fig. 6i and S6e), and deranged amounts of PAPPA (Fig. 6k) were also modified in inflamed adipocytes under treatment. Overall, these results reinforce the key relevance of impaired *PKP2* in inflamed/obese adipocytes, as by restoring this unique member of the armadillo-repeat family, it is possible to rewire E2F signaling towards the re-activation of the cell cycle, alleviating the burden of senescence in these cells.

**Fig. 5 | Gene expression measures in alternative *PKP2*-deficient fat cells.** Pictures **a**, **b** illustrate fat samples collection and handling to obtain the results that follow. Relevant information regarding these datasets is contained in supplementary Tables S5 (humans) and S6 (mice). **c** Control group ratio-relative gene expression measures (mean with SEM of the Log$_2$ fold-change) assessed in adipose tissue and ex vivo isolated adipose cells from heterozygous Pkp2$^{+/-}$ knockout and wild-type (Wt) mice (*n* = 5 and 4, respectively), the later taken as a reference. We have only labeled significant (*$p$ < 0.05; **$p$ < 0.01) and near-to-significant ($p$ < 0.2) comparisons to avoid figure overfilling. **d** Age and expression of biomarkers of interest in isolated SVCs from *PKP2*-mutated ACM cases and controls with an intact PKP2. Real-time PCR results are shown as the percent of variation with regard to the mean of the control group (i.e., non-PKP2-dependent ACM and non-ACM patients). *PKP2*

gene expression in these samples was undetectable (Cts >40). **e** Color in heat maps represent row $z$-scores calculated for each subject-derived cell type (by column) by subtracting the mean and then dividing by the standard deviation of each column. The value associated with each square stands for the mean of four biological replicates for each individual and cell. The numbers refer to the donor from which cells were retrieved, as depicted in the supplementary Table S5. **f** Boxplots show disaggregated raw data for genes of interest, including 75th to 25th percentiles with the median, and whiskers at maximum and minimum values. Data points include four biological replicates for each subject-derived cell type (i.e., *n* = 4 cells/group examined over four independent experiments). Statistical significance was assessed by a two-tailed Fisher's exact *t*-test. r.u. relative units; *$p$ < 0.05; **$p$ < 0.01. Source data are provided as a Source data file.

## Discussion

Cellular senescence is a state presenting with extended and generally irreversible cell cycle arrest[68]. Multiple stimuli, such as the physiological decay experienced during aging, including (but not limited to) DNA damage, chromatin remodeling, and telomere attrition[69], or the multivariable low-grade, chronic, and systemic body of responses that propel inflammatory issues[70,71], may trigger cellular senescence. This appears to prompt the loss of cellular key functions also in differentiated adipocytes[58,69]. Indeed, while the inflammatory processes associated with adipose tissue senescence seem to be pivotal to the accumulation of dysfunctional fat cell progenitors[72–74], a senescent profile is also observed within a spectrum of activated pro-inflammatory fates[75], comprising the anomalous activity of mature adipocytes[60]. Plakophilin-2 (PKP2) is a widespread desmosomal plaque protein with an apparent role in cell proliferation[76] that has been implicated in the non-canonical adipogenic conversion of deranged cardiac cells under defective PKP2[77]. Hitherto less attention has been given to the expression and function of PKP2 in adipose cells. In the current study, we show that PKP2 levels steadily increase during the adipogenic course, with expression levels reaching an apex in terminally differentiated adipocytes. Clinically, while omental PKP2 is more strongly associated with the ontogenetic presence of mesothelial progenitors within this fat depot, decreased PKP2 in subcutaneous adipose tissue is tightly related to obesity and inflammation, as shown by the systematic analysis of cross-sectional samples and longitudinal studies of weight loss. Accordingly, we identified impaired expression of *PKP2* in adipocytes upon treatment with macrophage-derived pro-inflammatory cytokines mimicking the microenvironment of obese adipose tissue. We demonstrated further that reduced expression of adipocyte-specific PKP2 has a consistent impact on fat cell function, while integrated multiomics (microarray and high-resolution mass spectrometry) in two loss-of-function approaches (viral vector and small interfering RNA-mediated) unveiled new functions for PKP2 in human adipocytes and fat cell progenitors, mostly characterized by the maintenance of cell cycle and E2F target gene signatures to avoid senescence under conditions of sustained inflammation. Conversely, restitution of PKP2 in human adipocytes provided an inner sight of the implications of the loss of PKP2 to realize the occurrence of deleterious events due to the anomaly of the impaired cell cycle in inflamed/obese adipocytes. A cartoon of the resulting model is presented in Fig. 7.

Muscle and bone cell lines can undergo adipogenesis and trans-differentiate into adipocytes when challenged with adipogenic stimuli[78], or by ectopically expressing PPARγ[79], which may led to diminished expression of anti-adipogenic canonical WNT signaling[80]. The specific knockdown of PKP2 has also been reported to suppress the E2F1 pathway to promote adipogenesis in HL-1 cardiomyocytes[77]. This converges with our results, pointing at the increased differentiation and enhanced acquisition of adipogenic traits in 3T3-L1 and human adipocyte precursors with impaired *PKP2*. Yet, cellular confluence (when cells in culture are in contact with one another) is a requirement for canonical adipogenesis[78]. Indeed, seeding

mesenchymal stem cells in a spread configuration, can lineage commit directly towards osteoblast fate, whereas they tend to become adipocytes when confluent[81]. Desmosomes are stable cell–cell adhesive junctions primarily found in epithelial and cardiac tissues, in which they improve the ability to withstand mechanical stress, while maintaining both intracellular signaling and direct intercellular interactions between neighbor cells[82]. Largely neglected in adipose tissue, scarce lines of evidence so far available suggest that junction dynamics may be of relevance to adapt adipocytes to their changing environment[83], which requires plasticity as well as a certain degree of stiffness against tension and mechanical strain[78]. Here, we provide findings linking the expression of PKP2 to the adipogenic course of canonical adipocytes, as a conversion of spindly fibroblasts to round adipocytes was characterized by increased PKP2 (but not other desmosome-associated genes), running together with a major remodeling of intracellular structures and the appearance of cytoplasmic lipid droplets. This underlies the so far unforeseen relevance of PKP2 in adipocytes, and renders expected impaired adipocyte function in situations, where PKP2 levels are compromised.

Importantly, our experiments revealed that expression of *PKP2* is dampened in inflamed adipocytes and obese subcutaneous adipose tissue, being restored in the latter upon weight loss. At the same time, we found that impaired *PKP2* in mature adipocytes modifies G1/S/G2/M-phase gene expression patterns suggestive of impaired E2F signal and metabolic disarrangement, with effect strength correlated to treatment length. Inflammatory cytokines may constitute the molecular basis of cell cycle arrest and the establishment of senescent-associated molecular patterns[84,85], contributing to the pathological consequences of chronic inflammation in obese adipose tissue[74]. Indeed, fat cells display gene and protein signatures indicative of an active cell cycle program, including the expression of G2M checkpoint genes and E2F transcription factors that may initiate either cell cycle arrest, proliferation, or death[58]. Recently, Qian Li and co-workers demonstrated that human adipocytes activate an endoreplicative program as part of the cellular response to their microenvironment, enabling further adaptation to the hyperplastic expansion of adipose tissue and hyperinsulinemia in patients with obesity[57]. Here we used terminally differentiated human adipocytes as a model system to understand how PKP2 can coordinate adipocyte functions and cell cycle, while simultaneously monitoring the impact of systemic exposure to the pro-inflammatory cytokines released by macrophages (MCM). By this approach, we observed that MCM treatment lasting 6 days (and thus, decreased expressions of *PKP2* in adipocytes, preventing the replenishment of this desmosomal protein) led to molecular patterns very close to those found upon the specific knockdown of *PKP2*. In these cell models, adipocytes presented attributes related to cellular senescence, including decreased amounts of the cell cycle driver cyclin D1, enhanced β-galactosidase activity and the subsequent release of H$_2$O$_2$, as well as changes in gene expression and protein biomarkers suggestive of impaired cell cycle, altered abundance of junctional proteins, and activated senescence, while the exogenous plasmid-based expression of human PKP2 seems to be sufficient to

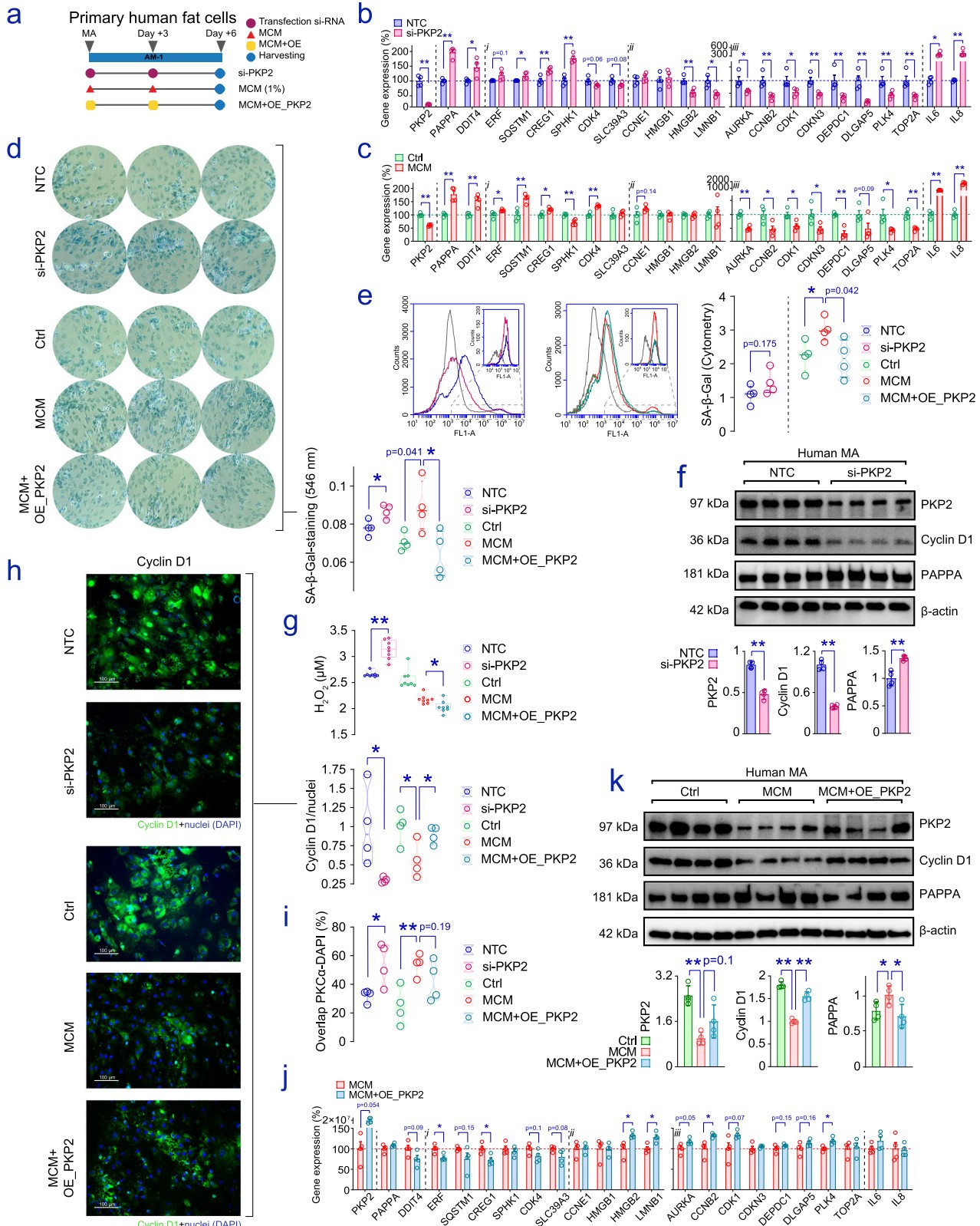

mitigate the chain of events leading to adipocyte failure. At the crossroad of these events, able to further the molecular pathophysiology of obesity, we found consistent variations affecting the appearance of nuclear protein kinase C alpha (PKCα) in adipocyte cultures in which PKP2 levels were modulated. These results are consistent with data from different models whereby the interaction between PKP2 and PKC isoenzymes, either directly or in conjunction

with other scaffolding proteins, was reported[1,86]. As a matter of fact, it is well recognized that calcium-activated, phospholipid-dependent PKC activity is involved in cellular functions ranging from growth control to intercellular adhesion, activation of cellular function, differentiation, and cell death[87,88]. The notion that intracellular translocation of activated PKCα regulates molecular events linking signal transduction pathways to the cell cycle machinery is also supported by

**Fig. 6 | Altered estates in inflamed adipocytes are softened upon *PKP2* restoration. a** A comprehensive pipeline was settled to sequentially validate altered cell cycle and enhanced senescence in MA under either defective PKP2 (si-*PKP2*) or enduring sustained exposure to MCM, also characterized by impaired *PKP2* gene expression. In the latter, we used plasmids to restore *PKP2* levels (MCM + OE_PKP2). Bar plots show the change for a set of genes previously identified as related to impaired *PKP2*, including those (i) directly or (ii) inversely related to senescence, and (iii) biomarkers downstream E2F signaling, also bound to cell cycle, in **b** si-PKP2 and **c** MCM-treated adipocytes. **d** SA-β-gal staining in MA under different conditions. Pictures show three representative images sorted out from four biological replicates in which measures of absorbance (546 nm) allowed the quantitative assessment plotted at the bottom right corner. **e** By using a BD Accuri C6 flow cytometer, the amounts of SA-β-gal positive adipocytes were assessed. Representative flow cytometric curves are provided for each treatment. The gray line shows cell distribution in unmarked control cells. The gating strategy is exemplified in supplementary Fig. S6c, d. The bean plot on the right indicates the amounts of SA-β-gal positive adipocytes for each condition (*n* = 4 biological replicates/condition). **f** Western blot analysis for PKP2, Cyclin D1 and PAPPA in non-targeting controls (NTC) and si-PKP2-treated MA. The bar plots below indicate β-actin-normalized relative amounts of each protein. **g** The box plot (median values plus 25th and 75th percentiles, and whiskers bound to maximum and minimum

values in each group) shows concentrations of hydrogen peroxide ($H_2O_2$) measured in conditioned media supernatants collected from cell cultures of either NTC or si-PKP2 adipocytes (left boxes), or MCM-inflamed adipose cells (right boxes) containing a plasmid with an empty cassette (reference) or coding for *PKP2* (*n* = 8 biological replicates/condition). **h** Representative 20x immunofluorescent images (the scale bars denote 100 μm length) showing cell cycle driver Cyclin D1 signal in adipocyte cultures. The bean plot at the right-hand shows the Cyclin D1 signal after correcting for blue fluorescent stain DAPI in four biological replicates/conditions. Below, **i** the percentage of nuclei showing Protein Kinase C alpha (PKCα) immunofluorescence staining in adipocyte cultures following the treatments depicted above. Representative 20x immunofluorescent images are provided in Fig. S6e (scale bars of 100 μm), and an example of image handling and data collection is shown in Fig. S6f. **j** Gene expression patterns and **k** Western blot analysis of PKP2, PAPPA, and Cyclin D1 in cultures of adipocytes maintained undisturbed (Ctrl) and inflamed (MCM) adipocytes in which we used plasmids to restore *PKP2* levels (MCM + OE_PKP2). Ctrl and MCM groups were treated with the negative empty control vector for pReceiver-M90 and transfection reagent, as per proper control conditions. Bar plots show results assessed in four biological replicates/conditions, and the mean ± SEM Statistical significance was assessed by two-tailed Fisher's exact *t*-test. *$p < 0.05$; **$p < 0.01$. Source data are provided as a Source data file.

compelling evidence readily available in the literature[89,90]. The underlying mechanism in adipocyte-specific *PKP2* deficiency potentially comprises the translocation of nuclear PKCα, and supports the hypothesis that PKP2 prevents aberrant molecular patterns in adipocytes by locally tying PKCα to the cytoplasm, thus preventing changes in the distribution of active PKCα and the phosphorylation of PKC nuclear substrates related to the synthesis of proteins involved in cytoskeletal dynamics, junction assembly, cell cycle, and adipocyte differentiation and maintenance.

Senescent adipocytes may have profound clinical consequences because of their major role in the metabolic control exercised by adipose tissue, and thus senolytic treatments aimed at removing senescent cells from fat depots have gathered much interest as a therapeutic option[91,92]. Our functional and mechanistic results, though constrained by the limitations inherent to cell models, imply possible new avenues of research on the pathogenesis and potential treatment of adipocyte malfunction in the context of obesity, inflammation, fat cell hypertrophy, and impaired metabolism. It is tempting to propose that our findings may represent a first step toward a better understanding of how the size of adipose tissues and inflammation occurring in patients with obesity may regulate the expression of genes required to maintain adipocyte well-being. Manipulating *PKP2* expressions to restore adipose cellular makeup may therefore constitute an attractive drug-development target to combat obesity-associated inflammatory issues and metabolic complications.

## Methods
### Human cell cultures
Cryopreserved preadipocytes (PA) were commercially available (Zen-Bio, Inc.) and were obtained from the subcutaneous (SC) and omental (OM) adipose tissues (SP-F-2 and OM-F-2, respectively) of one female donor of 34 yrs and a body mass index (BMI) of 26.3 kg/m². SC adipocyte progenitor cells from one female donor without (BMI <25 kg/m²) and another one with (BMI >30 kg/m²) obesity (SP-F-1 and SP-F-3, respectively), both of approximately the same age (35–40 yrs), were also plated in culture and stimulated with adipogenic conditions, as previously explained[93]. Briefly, confluent preadipocytes in the pre-adipocyte medium (PM-1) were incubated with adipocyte differentiation medium (DM-2) for 7 days. This media is composed of DMEM / Ham's F-12 (1:1), HEPES, FBS, biotin, pantothenate, insulin, dexamethasone, IBMX, PPARγ agonist, penicillin, streptomycin, and amphotericin B. Thereafter, differentiating adipocytes were maintained in adipocyte maintenance medium (AM-1), formulated as the DM-2 media but without dexamethasone, IBMX, and PPARγ agonist

(Fig. 1a). Several other vials SP-F-2 were purchased, cultured, differentiated and treated as explained below. Two weeks after initializing adipogenesis (14th day), differentiated cells appeared rounded with large lipid droplets apparent in the cytoplasm, and were then considered mature adipocytes (MA). The human monocyte cell line THP-1 (ATCC, TIB-202) was grown at 37 °C in a humidified 5% $CO_2$ and 95 °C air atmosphere in RPMI 1640 medium (Gibco, 21875-034) containing 10% FBS, 5 mM glucose (Sigma, G8644), 2 mM L-glutamine, 50 mg/ml Gentamicin (Gibco, 15710-064), and 20 mM HEPES. The pro-inflammatory type 1 macrophage-like state (M1) was induced with 0.162 mM phorbol 12-myristate 13-acetate (Merck, P1585) for 24 h. Then, plastic-adherent M1 macrophages were washed with PBS and incubated with fresh medium without PMA for 24 h. Finally, M1 cells were cultured with fresh medium or fresh medium containing 10 ng/ml lipopolysaccharide (Sigma, L4516). After 24 h, the media were collected and centrifuged at 400×*g* for 5 min, diluted in adipocyte medium, and used as a macrophage (MM) or macrophage LPS-conditioned (MCM) media to mirror in cultures of MA the pro-inflammatory milieu of an obese adipose tissue (Fig. 1d).

### 3T3-L1 cells
The embryonic mouse 3T3-L1 fibroblast cell line (ATCC, CL-173) was cultured in Dulbecco's Modified Eagle's Medium (DMEM) containing 4.5 g/l glucose (Gibco, 21013-024), 10% FBS, 100 U/ml penicillin, and 100 μg/ml streptomycin. Three days after reaching confluence, adipogenic differentiation was induced by a cocktail of 1 μg/ml insulin (Sigma, I3536), 0.25 μM dexamethasone (Sigma, D1756), and 500 μM isobutyl methylxanthine (Sigma, I7018). This mixture was added to the media for 4 days, followed by 4 days of basal cell culture media plus 1 μg/ml insulin to conclude proper 3T3-L1 cell evolution into differentiated lipid-containing MA (Fig. 1f). Non-differentiated 3T3-L1 cells were cultured in parallel but maintained in basal grow media. Intracellular lipid accumulation in adipocyte cultures was measured using Oil Red O staining (Abcam, ab150678). For immunofluorescence, 3T3-L1 cells were plated, grown, and differentiated on collagen-coated glass coverslips. Then, cell monolayers were rinsed with PBS and fixed with 4% paraformaldehyde at RT in a 0.01 M PBS buffer (pH 7.4) for 10 min, before being incubated for 3 min with 0.1% Triton X-100 (Sigma, T8787), and treated with a blocking solution (10% goat serum (Sigma, NS02L), and 1% bovine serum albumin (Sigma, A2153) in PBS) for 1 h. Then, 1:200 dilution of primary mouse monoclonal antibody against PKP2 (SCBT, clone C1, sc-393711) was applied for 2 hrs, followed by 1:200 dilution of Alexa Fluor 546 Goat anti-rabbit (Invitrogen) for 1 h, and counterstained with 1 μg/ml DAPI (Sigma, D9564) for 5 min

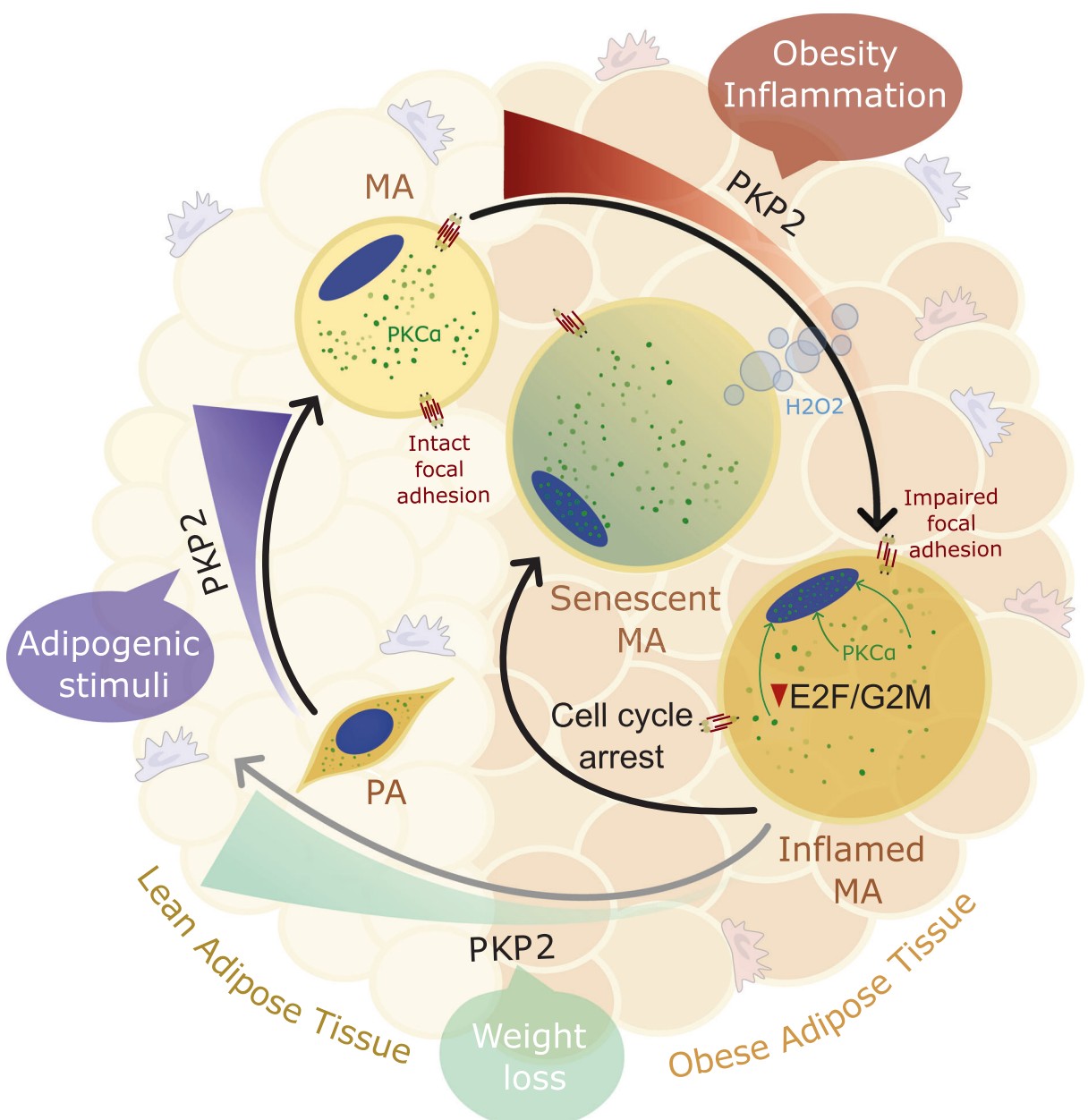

**Fig. 7 | PKP2 dynamics in adipocytes.** Our data show that Plakophilin-2 (PKP2) is increased during adipogenesis, with expression levels reaching an apex in terminally differentiated adipocytes. Conversely, sustained low-grade inflammatory conditions affecting adipose tissue in individuals with obesity may promote a deficit in adipocyte-specific PKP2 that disrupts the cell cycle transcription program and its conservative function, leading to activated adipocyte senescence. This phenomenon is potentially driven by the nuclear translocation of protein kinase Cα (PKCα), a scaffold regulator linking differentiation, signal transduction, and cell-cycle machinery. We also demonstrate that, in parallel to transcriptional reprogramming and changes affecting the amount of proteins related to cell cycle, cell-to-cell interaction, and cellular stress in adipocytes, impaired *PKP2* may conduct the enhanced release of $H_2O_2$, worsening the deleterious consequences of an inflammatory milieu. We report that restoring PKP2 in adipose tissue (by significant weight loss) and inflamed adipocytes (by plasmid-based genetic tools) normalizes the subcellular location of PKCα, rewires E2F signaling towards re-activation of cell cycle, reduces the extrusion of $H_2O_2$, and decreases adipocyte senescence, suggesting that recovery of adipocyte-specific PKP2 can be envisaged as a therapeutic goal for patients with obesity and related diseases.

at RT. Slides were washed with cold PBS and mounted with Dako fluorescent mounting medium (Agilent, CS70330-2). Images were captured on a ZEISS Axiovert 200 inverted microscope equipped with an AxioCam camera system.

**Loss-of-function approaches**
Lentiviral particles containing short hairpin (sh)-RNAs, and small interfering (si)-RNAs against PKP2-coding genes were used in mouse and/or human MA or in adipocyte progenitor cells. Permanent silencing was achieved in 3T3-L1 fibroblasts using *Pkp2*-targeted (sc-43183-V)

and scrambled non-silencing (NS) control (sc-108080) lentiviral particles purchased from Santa Cruz Biotechnology, together with 10 μg/ml polybrene (Sigma, TR-1003). 3T3-L1 preadipocytes harboring sh-RNA cassettes were enriched by means of 3 μg/ml puromycin dihydrochloride (Sigma, P8833) 48 h post-infection. The same procedure was applied to fully-differentiated human adipocytes (sc-43182-V), before inducing the adipogenic conversion. To rule out any altered states mediated through the combination of lentiviral infection and puromycin selection, human PA and lipid-filled MA were also cultured in the ectopic presence of 100 nM of si-RNAs targeting *PKP2* or a

non-targeting control (NTC). The si-RNAs used during this research were against human *PKP2* (Sigma, SASI_HS01_0014063), and showed the sequences 5′-CAGAUUACCAGCCAGAUGA[dT][dT]-3′ and 5′-UCAU-CUGGCUGGUAAUCUG[dT][dT]-3′. In parallel, MISSION® siRNA Universal Negative Control #1 (Sigma, SIC001), combined 1:3 with Lipofectamine RNAiMAX (Invitrogen, 13778-075), was used as a control for the indicated periods of time (see in Figs. 3a, 4a, 6a assays' pipelines). Transduction efficiency, and the knockdown of *Pkp2* in sh-*Pkp2* 3T3-L1 cells was monitored for GFP expression, and verified by quantification of the target transcript (real-time PCR) and protein levels (Western blot) in murine and human adipocytes and precursor cells.

### Plasmid-based PKP2 expression

The expression-ready human *PKP2* cDNA ORF clone (#EX-Y4017-M90), and the negative empty control vector for pReceiver-M90 (#EX-NEG-M90), were obtained from Genecopoeia (Tebu-bio, Spain) and transfected (0.25 μg/cm²) into MA challenged with 1% MCM for 6 days. The pReceiver-M90 vector incorporates the human cytomegalovirus (CMV) promoter and SV40-eGFP-IRES-Puromycin Tag. Successful transfection of human adipocytes was achieved by Fugene® Transfection Reagent (Promega, WI, USA) at a ratio of 1:3, following the manufacturer's instructions. Transfection efficiency was checked by measuring PKP2 levels in harvested cells (real-time PCR and Western blot).

### Human fat samples

Adipose tissue was obtained by biopsy at the mesogastric level before and ~2 years after bariatric surgery[31]. Additional fat samples were obtained from 24 age-matched women with BMI <30 kg/m² and following elective surgical procedures (e.g., cholecystectomy and surgery of abdominal hernia). An independent sample of ~220 subjects aged between 20 and 80 yrs (47 ± 11 yrs; 22% men), including patients at the extremes of the weight continuum (BMI = 35 ± 11.2 kg/m²; 50% obesity), was enrolled for a comprehensive assessment of the relationship between SC and OM adipose *PKP2* and clinical and biochemical measures. Three hundred and fourteen samples were obtained from OM (n = 131) and SC (n = 183) depots of fat (92 paired samples). This cross-sectional sample comprised the general population following elective surgical procedures, and obese and diabetic outpatient clinics who showed stable metabolic control in the previous 6 months, as defined by constant values for weight and glycosylated hemoglobin. Baseline studies included a standardized questionnaire, physical examination, and common laboratory tests. Height and weight were measured with the participant in light clothing and without shoes by trained personnel using calibrated scales and a wall-mounted stadiometer, respectively. BMI was calculated by dividing weight in kilograms by the square of the height in meters (kg/m²). Impaired glucose tolerance (IGT) was defined as fasting glucose levels above 110 mg per dL (5.6 to 6.9 mmol per L) and/or two-hour glucose levels of 140 to 199 mg per dL (7.8 to 11 mmol per L) on a 75-g oral glucose tolerance test. The characteristics of participants in cohorts 1 and 2 are described in Table S1 and S2, respectively. The fat tissue was selected from the greater stomach curvature and sectioned at its lowest point. All macrobiospies were performed without electrical and/or heating devices, introduced in a sterile 15 ml container and immediately flash frozen in indirect contact with liquid nitrogen. Then, fat samples were stored at −80 °C until the next procedure. For the analysis of PKP2 in fat cell populations, isolated stromal vascular cells (SVC) and MA were obtained from ~5 g of OM and SC adipose tissue. Biopsies from 18 women with morbid obesity were finely minced, enzymatically disaggregated with a collagenase solution (Sigma, C0130) at 37 °C for 20 min, centrifuged, filtrated, and isolated according to a modification of the Bunnell method[94]. The nature of the buoyancy adipocytes allows them to float to the surface. Then, the floating adipocyte layer and the pelleted of SVC were collected, washed, processed and studied separately (see in Fig. 2i). All subjects were white and gave written informed consent. Samples and data from participants included in this study were provided by the FATBANK platform, promoted by the CIBEROBN, and coordinated by the IDIBGI Biobank (Biobanc IDIBGI, B.0000872). All samples were processed following standard operating procedures with the appropriate approval of the Ethics, External Scientific, and FATBANK Internal Scientific Committees. These Ethics Committees approved the study procedures and written informed consent was obtained from all participants before inclusion in the study.

### Human stromal fat cell isolation and culture

This study complies with the Declaration of Helsinki and was approved by "Istituto Europeo di Oncologia e Centro Cardiologico Monzino IRCCS" Ethics Committee (CCM1072-03/07/2019). Subcutaneous adipose tissue biopsies were obtained from patients undergoing the implantable cardioverter defibrillator procedure after signing an informed consent. We analysed five patients carrying a pathogenic mutation in *PKP2*, and five non-*PKP2*-mutated patients (Table S5). Human subcutaneous fat-derived stromal cells were isolated as previously described in ref. 54. Briefly, bioptic samples were washed with PBS, minced using sterile scissors, and incubated in IMDM (Invitrogen, #21980-032) containing 3 mg/ml collagenase NB4 (Serva, DS17454) at 37 °C for 1.5 h under continuous agitation. The digested solution was then at 400×*g* for 10 min. The obtained pellet was snap-frozen and stored at −80 °C or resuspended in IMDM supplemented with 20% FBS, 10,000 U/ml Penicillin (Invitrogen, #15140-130), 10 mg/ml Streptomycin (Invitrogen, #11860-038), 20 mmol/l L-glutamine (Sigma-Aldrich, G7513), and 10 ng/ml basic fibroblast growth factor (R&D Systems, #3718-FB). The cells were seeded onto uncoated dishes and non-adherent cells were removed after 24 h. After two passages in culture, cells were removed and frozen in FBS 10% DMSO. When required, these cells were seeded in 24-well plates with preadipocytes media (DMEM/F-12 10% FBS with antibiotics), led to proliferate until reaching confluence, and differentiated as explained in ref. 93.

### Pkp2⁺/⁻ mouse samples

C57Bl/6 *Pkp2* heterozygous knockout mice ($Pkp2^{+/-}$) were produced by Prof. Walter Birchmeier's lab, as described in ref. 2. For our current purposes, nine 3-to-5-month-old mice were studied (four males and five females), including five $Pkp2^{+/-}$ mice, and four wild-type (Wt) siblings used as control (Table S6). Mice were housed in a room with controlled temperature (~20 °C) and humidity (~50%), in a 12:12 light/dark cycle (6 a.m.–6 p.m.), and had free access to water and a standard chow diet with 15 Kcal% fat (OpenSource Diets, D11112201). Experiments were authorized by the Italian Ministry of Health, protocol no. 249/2020-PR. Animals were terminated by carbon dioxide inhalation, the external surface was sterilized with 70% ethanol previous surgery, and subcutaneous fat samples were collected. Briefly, the mouse was placed on its back in the supine position, and a lateral incision was made on the abdomen below the diaphragm. From the midpoint of the first incision, an additional cut was made along the midline toward the rectum. The subcutaneous fat pads from both flanks of the mouse were dissected, and a small piece was snap-frozen and stored at −80 °C for RNA extraction. The residual part was processed for isolation of adipocytes and SVC as follows: fat samples were minced using sterile scissors and digested in DMEM/F-12 (Invitrogen, #11320-033) containing 4 mg/ml of Collagenase type 2 (Worthington, LS005275) in a shaker-incubator at 37 °C for approximately 40 min. Digested material was filtered through a 100 μm strainer, washed with DMEM/F-12 + 10% FBS, and spun at 50×*g* for 3 min in order to discard the undigested samples. The infranatant was removed and floating adipocytes were washed again. SVC were isolated from the first infranatant by centrifugation at 400×*g* for 10 min. Both adipocytes and SVC were snap-frozen and stored at −80 °C previous further analyses.

## Histology

Histological analyses were performed using SC and OM adipose tissues from the same donors ($n = 3$). Samples were collected, snap-frozen, and stored at $-80\,°C$ until processed. Before sectioning, deep-frozen adipose tissue was embedded in Tissue-Tek O.C.T. compound (Sakura Finetek, 4583), and placed in a cryostat for stabilization at $-25\,°C$. About $18\,\mu m$ coronal slides were cryosectioned and mounted in Superfrost™ Plus Microscope pre-coated slides (Fisherbrand, 22-037-246). Adipose tissue was first let dry at RT for 2 h and fixed using 4% paraformaldehyde (pH 7.4) for 15 min, then washed three times with PBS before staining. After fixation and rinsing, the slides were stained with Mayer's hematoxylin solution (Sigma, MHS32-1L) for 30 s. Then, the excess hematoxylin reagent was removed, and samples were partially dehydrated with 70 and 95% ethanol before adding Eosin Y staining reagent (Sigma, E4009). Finally, the dehydration was completed with a sequence of 95%, 100% ethanol, and xylene and mounted using the Fisher Chemical™ Permount™ mounting media.

## Immunofluorescence

After complete paraformaldehyde (PFA) removal, tissue was permeabilized in PBS 0.05% Triton X-100 and blocked with 10% Normal Goat Serum (Abcam, ab7481) for 1 h. Then, samples were incubated with 1:250 primary mouse monoclonal antibody against PKP2 (SCBT, clone C1, sc-393711) at $4\,°C$. The day after, tissue was washed in PBS and incubated at 1:500 with an Alexa Fluor™ 594 secondary antibody (Abcam, ab150116) for 1 h at RT. Then, sections were washed, Hoechst Stain solution (Sigma, 23491-45-4) was added, and the immunofluorescent evaluation was carried out in a Leica AF6000 microscope (Leica Microsystems). For experiments in vitro, adipocytes were cultured, differentiated, and treated on coverslips coated with $20\,\mu g/mL$ fibronectin (Innoprot, #P8248), then fixed (4% PFA), permeabilized (0.1% saponin) and blocked (1% BSA). Fixed cells were incubated with primary antibodies against Cyclin D1 (Cell Signaling, #55506, clone E3P5S) or PKCα (#2056), followed by incubation with goat anti-Rabbit IgG (H + L) Highly Cross-Adsorbed Secondary Antibody Alexa Fluor™ 488 (Invitrogen, #A-11034). Nuclei were stained with DAPI (Thermo, #62248). Coverslips were mounted on slides with Vectashield mounting media (Vector Laboratories, #H1000), and pictures were taken by means of a Nikon Eclipse 50i Confocal Microscope, with the 20x objective of 1.4 NA. Nikon's Flagship NIS-Elements Package software (version 4.2) was used for image analysis. Fluorescein 5-isothiocyanate (FITC) staining in nuclei was assessed by FIJI software (https://fiji.sc/) software (v1.53t). Channels were splinted and threshold adjusted, and the area of each channel was selected. Then, the areas of each channel and the overlap was measured, and the percent area overlap/total nuclei area was calculated.

## Real-time PCR

Total RNA was obtained from cell cultures and human samples using the RNeasy Mini Kit (QIAgen, 74104). The integrity was checked by means of an Agilent Bioanalyzer System. RNA was quantified in a NanoDrop 1000 Spectrophotometer (Thermo Scientific), and $3\,\mu g$ were retro-transcribed into cDNA using the High Capacity cDNA Archive Kit (Applied Biosystems, 4368814). The reverse transcription was conducted using the following cycle parameters: $25\,°C$ for 10 min, followed by 2 h at $37\,°C$, and a final inactivation step of 5 min at $85\,°C$. All cDNA samples were applied in dilution of 1:10 to obtain results within the range of the standards. Commercially available TaqMan hydrolysis probes (Applied Biosystems) and SYBR Green I, with forward/reverse paired primers (KiCqStart® SYBR® Green Primers; Sigma, KSPQ12012), were used to assess the expression of gene candidates in a Light Cycler 480 II sequence detection system (Roche Diagnostics). The following thermocycler conditions were used: $50\,°C$ for 2 min and $95\,°C$ for 10 min, followed by 40 cycles for 30 s at $95\,°C$, 45 s at $60\,°C$ and $72\,°C$ for 30 s. The gene expression levels were normalized to the reference gene transcript *cyclophilin A*. The normalized fold expression was obtained using the $2^{-\Delta\Delta Ct}$ method, and results are expressed as the normalized fold expression for each gene as compared to untreated/ reference control cells. Replicates and positive and negative controls were included in all reactions. TaqMan hydrolysis assays and paired SYBR Green primer sequences are listed in Table S7.

## Microarrays

Complete transcriptomic data supporting the findings of this study is MIAME compliant and available in the community-endorsed repository Gene Expression Omnibus (GEO), with accession code numbers GSE182231 and GSE182229. Gene expression transcriptomes were obtained using Genechip® Affymetrix technology. All procedures, including sample processing, hybridization, developing, and chip scanning, were performed following the manufacturers' protocol for Human Gene 2.0 ST array, Ambion® WT expression Kit, and the Genechip® WT Terminal Labeling kit. R programming (Version 3.6.0) was used, together with different packages from Bioconductor (Bioc 3.10)[95] and the Comprehensive R Archive Network (CRAN; https://cran.r-project.org). After quality control of raw data, samples were background corrected, quantile-normalized, and summarized to a gene level using the robust multi-chip average (RMA)[96] obtaining a total of 20,893 transcripts. Then, clustering methods were applied. An empirical Bayes moderated t-statistics model (LIMMA)[97] was built to detect differentially expressed genes between the studied conditions. Correction for multiple comparisons was performed using false discovery rate (FDR)[98], and adjusted $p$ values were obtained. Clariom-S_Human array CEL files were analysed for differential gene expression using Transcriptome Analysis Console (TAC 4.0) software (Affymetrix) to explore changes affecting transcriptome profiles between experimental groups. Summarization of gene level was performed using the SST-RMA method and genome version hg38 (*Homo Sapiens*). The background was adjusted by a quantile normalization and a log2 data transformation. The significant genes were ranked by ANOVA $p$ value <0.05. Heat maps represent $z$-scores based on gene counts for each sample using the package Bioinfokit (2.0.8) for Python. Gene Set Enrichment Analysis (GSEA) was performed with lists of probes showing significant changes with regard to respective controls. The FDR $q$ value score of 0.05 was set as a threshold. A pre-ranked metric analysis was performed using the log-fold-change following the guidelines set by the Broad Institute (https://software.broadinstitute.org/gsea/doc/GSEAUserGuideFrame.html). Three collections of gene sets were evaluated: Hallmark, C2 (containing all curated gene sets), and C5, which includes the gene ontology (GO) biological process. Pathways diagrams were performed using the Enrichment Map application in Cytoscape (v.3.7.2)[99]. The lists of probes were also uploaded into the Ingenuity Pathways Analysis (IPA) 8.7 software (Ingenuity System, Inc., http://www.ingenuity.com), and the 'Core Analysis' function included in IPA was used to interpret the results in the context of biological pathways and networks. In addition, we used Metascape (https://www.metascape.org)[45] to create an interactome analysis to complement IPA results. Finally, the relative abundance and changes affecting transcripts coding for Plakophilin 1 to 4 in the human adipose tissue upon weight loss[31–34] and in vitro cultured or ex vivo isolated adipocytes[19,20,23] and SVC[35] was achieved by re-analysing independent transcriptomes downloaded from public repositories.

## Proteomics

Purified peptides from fat cell cultures ($n = 4$ biological replicates/group) were analysed by liquid chromatography-tandem mass spectrometry (MS) carried out on a Bruker Daltonics timsTOF Pro instrument connected to a Bruker Daltonics nanoElute nanoflow UHPLC. Sample prep started from protein lysates of preadipocytes and mature adipocytes. Cells (three 24-wells per replicate) were collected in 4% sodium lauroyl sarcosinate (SLS) in 100 mM Tris-HCl

(pH-7.5) solution and boiled for 10 min at 90 °C in a ThermoMixer (1800 rpm). Subsequently, supernatants were collected by 13,000×$g$ centrifugation at 4 °C for 15 min, and protein concentration was estimated by Lowry Assay. Prior to protein digestion, DNA was removed by two cycles of incubation at 90 °C in a ThermoMixer (10 min, 1800 rpm) followed by 1 min sonication. Approximately 20 μg of protein (approx. 35 μl of solution) were taken for further sample processing. Samples were reduced (addition of 3 μl of 400 mM dithiothreitol to a final concentration of 10 mM and incubation at 95 °C for 10 min), followed by alkylation (additions of 3 μl 550 mM iodoacetamide to a final concentration of 13 mM and incubation for 30 min at 25 °C). All samples were diluted 7x with 50 mM TEAB buffer to allow trypsin digestion. Trypsin was added to a final amount of 0.6 μg. Digestion was carried out for 16 h at 37 °C and was stopped by the addition of TFA to a final concentration of 1.5%. Precipitating SLS was removed by centrifugation. Peptides were purified using solid phase extraction on C18 microspin columns according to the manufacturer's instructions (Macherey-Nagel, Germany). Purified peptides were first dried, then resuspended in 50 μL of 0.1% TFA. Peptide concentration was estimated using the Pierce Fluorimetric Peptide Assay, and sample volumes were adjusted to achieve equal concentrations. Subsequently, 300 ng of peptides were loaded onto a C18 precolumn (Thermo Trap Cartridge 5 mm, μ-Precolum TM Cartridge / PepMap TM C18, Thermo Scientific) and eluted in the backflush mode with a gradient from 98% solvent A (0.15% formic acid) and 2% solvent B (99.85% acetonitrile and 0.15% formic acid) to 17% solvent B over 36 min, continued from 17 to 25% of solvent B for another 18 min, then from 25 to 35% of solvent B for 6 min over a reverse-phase high-performance liquid chromatography (HPLC) separation column (Aurora Series Emitter Column with CSI fitting, C18, 1.6 μm, 75 μm × 25 cm, Ion Optics) with a flow rate of 400 nL/min. The outlet of the analytical column was directly coupled to the MS instrument using a captive spray fitting. Data were acquired by means of a data-independent acquisition (DIA) paradigm using a default method provided by the manufacturer. In short, spectra were acquired with a fixed resolution of 45,000 and mass range from 100 to 1700 m/z for the precursor ion spectra and a1/k0 range from 0.6 to 1.6 Vs/cm$^2$ with 100 ms ramp time for ion mobility, followed by DIA scans with 21 fixed DIA windows of 25 m/z width, ranging from 487.5 to 1,012.5 m/z. Peptide spectrum matching and label-free quantitation were subsequently performed using DIA-NN[52] and a library-free search against the Human Uniprot.org database (20,407 Swiss-Prot entries, April 2023). In brief, output was filtered to a 1% false discovery rate on the precursor level. Deep learning was used to generate an in silico spectral library for library-free search. Fragment m/z was set to a minimum of 200 and a maximum of 1800. In silico peptide generation allowed for N-terminal methionine excision, tryptic cleavage following K*, R*, a maximum of one missed cleavage, as well as a peptide length requirement of seven amino acid minimum and a maximum of 30. Cysteine carbamidomethylation was included as a fixed modification and methionine oxidation (maximum of two) as a variable modification. Precursor masses from 300 to 1800 m/z and charge states one to four were considered. DIA-NN was instructed to optimize mass accuracy separately for each acquisition analysed and protein sample matrices were filtered using a run-specific protein $q$ value. Downstream data processing and statistical analysis were carried out by the Autonomics package developed in-house (version 1.1.7.14; ref: https://doi.org/10.18129/B9.bioc.autonomics). Proteins with a $q$ value of <0.01 were included for further analysis. MaxLFQ[100] values were used for quantitation and missing values were imputed. DIA-NN spectral identification software initially identified 8214 protein groups. All intensities and maxLFQ values containing only 1 precursor (Np) per sample were exchanged by NA for that particular sample. After dropping 52 without replication (within subgroup), and filtering out 1684 proteins with less than

two peptides identified, 6478 protein groups were retained for further analysis. Differential abundance of protein groups was evaluated by Autonomics employing Bayesian moderated t-test as implemented by limma[97]. Functional enrichment analysis of DAPs was performed using MetaboAnalyst 5.0[101] and g:Profiler (version e109_eg56_p17_1d3191d) with g:SCS multiple testing correction method applying a significance threshold of 0.05[102]. The full list of settings can be found in the "report.log.txt" uploaded along with the mass spectrometry raw data to the ProteomeXchange Consortium, with dataset identifier PXD042110, and via the MassIVE partner repository (https://massive.ucsd.edu/ProteoSAFe/static/massive.jsp; https://doi.org/10.25345/C55M62H7G).

## Western blots
Proteins were extracted from cell monolayers directly in radio-immnuno precipitation assay (RIPA) buffer (0.1% SDS, 0.5% sodium deoxycholate, 1% Nonidet P-40, 150 mM NaCl, and 50 mM Tris-HCl, pH 8.0) supplemented with proteases inhibitors (1 mM phenylmethylsulfonyl fluoride, 2 g/mL aprotinin, and 2 g/mL leupeptin). Cellular debris were eliminated by centrifugation of the solubilized samples at 13,000×$g$ for 60 min at 4 °C, recovering the soluble fraction and avoiding the non-homogenized material at the bottom of the centrifuge tube. Protein concentrations were determined using a protein assay kit (Life Technologies, 22660/22663), and equal amounts of proteins were subjected to SDS-PAGE and transferred to nitrocellulose membranes (Whatman, 10600001). PKP2 was measured by western blot as follows: RIPA protein extracts (20 μg) were resolved on 10% SDS-PAGE and transferred onto Hybond ECL nitrocellulose membranes by conventional western blot procedures. Membranes were blocked with 5% (w/v) BSA in TBS buffer with 0.1% Tween 20. Immunoblotting was performed with 1:100 mouse monoclonal anti-PKP2 (SCBT, clone C1, sc-393711) and 1:2000 mouse monoclonal anti-β-actin (SCBT, sc-47778, clone C4) antibodies. Other antibodies used in this study include hFAB™ Rhodamine Anti-Tubulin Primary Antibody (#12004166) from Bio-Rad, rabbit anti-FAS (sc-20140) from SCBT, anti-PAPPA (AF2487) from R&D Systems, and anti-phospho-p38 MAPK (Thr180/Tyr182) (#9215), anti-p38 MAPK (#9212), anti-phospho-Akt (Ser473) (#9271), anti-Akt (#9272) and anti-Cyclin D1 (#2978, clone 92G2C8) from Cell Signaling Technologies. Following incubation with the primary antibody diluted 1:1000, the membranes were washed in TBS with 0.1% Tween 20 and incubated with the appropriate 1:2000 IgG HRP-conjugated secondary antibody (i.e., polyclonal rabbit anti-mouse (#P0260) from Dako, anti-rabbit IgG HRP-linked antibody (#7074) provided by Cell Signaling Technology, or goat IgG HRP-conjugated antibody (#HAF017) from R&D Systems). Immunoreactive bands were visualized with an ECL-plus reagent kit (GE Healthcare). Optical densities of the immunoreactive bands were measured using ImageJ analysis software (v.1.53k).

## SA-β-gal
Senescence-associated β-galactosidase (SA-β-gal) activity in human adipocytes was quantified by means of a commercially available colorimetric assay (Cell signaling, 9860). Human adipocytes were rinsed with PBS and fixed for 15 min at RT. After washing, β-Galactosidase-Staining Solution was added following the manufacturer's instructions, and cells were left to incubate at 37 °C overnight. Stained human adipocytes were then ready to be checked under a Nikon Eclipse Ts2 inverted microscope for the development of blue color. The SA-β-gal activity was also quantified by means of flow cytometry and The Senescence B-Galactosidase Activity Assay Kit (#35302, CST). Briefly, cells were incubated with Bafilomycin A1 (#54645) at 100 nM for 1 h at 37 °C. Then, SA-β-Gal Fluorescent Substrate (#38154) was added, mixed gently, and incubated at 37 °C for 3 h. Cells were washed three times with 1x PBS before being trypsinized and inactivated with PBS containing 2% FBS. Following the above methods, a flow cytometer BD

Accuri™ C6 Plus Flow Cytometer (BD Biosciences) and the BD Accuri™ C6 software (v1.0.264.21) were employed to evaluate the percentage of SA-β-gal-positive cells. The gating strategy is exemplified in supplementary Fig. S6c, d, where the selected adipocyte population is shown for all conditions.

### Statistical analyses

For cell and animal studies, data were expressed as mean ± standard error of mean (SEM). For human study analyses, results are presented in bean plots, and provided in supplementary tables as mean ± standard deviation (SD). Statistical significance between groups was determined using two-tailed paired and Fisher's exact t-test (two groups), or one-way analysis of variance (ANOVA) followed by Tukey's honestly significant difference (HSD) post hoc test when comparing multiple groups (3 and above). Statistical significance was assessed in time-course experiments by adjusted ANOVA after applying Šidák's corrections to repeated measures. Spearman's correlation analyses were performed between OM and SC *PKP2* and other quantitative variables, and by stepwise multiple linear regression analyses after adjusting for potential confounders. Statistical analyses were performed with the SPSS statistical software (IBM, Inc., Chicago, IL), GraphPad Prism Version 5.0 (GraphPad Software Inc., San Diego, CA), and/or *R* Statistical Software (http://www.r-project.org/).

### Reporting summary

Further information on research design is available in the Nature Portfolio Reporting Summary linked to this article.

## Data availability

The data that support the findings of this study has been deposited in public data repositories, and/or is presented as Supplementary Information with the manuscript. The microarray data generated have been deposited in the GEO repository under accession codes GSE182231 and GSE182229. The full list of mass spectrometry settings has been uploaded along with proteomics raw data to the ProteomeXchange Consortium PXD042110. Additional datasets used during this research are publicly available in the GEO repository under the following accession codes: GSE53378[31], GSE199063[32], GSE95640[33], GSE77532[19], GSE186519[20], GSE14312[23], and GSE135776[35]. The source data underlying the plots contained in this article is also provided as a Source Data file and/or was uploaded to Figshare (https://doi.org/10.6084/m9.figshare.23257538). The raw data deposited in Figshare includes differentially expressed (DE) genes (FDR *p* value <0.05) in our microarrays, and untargeted proteomics data, as assessed in preadipocytes (PA) and mature adipocytes (MA), including the pathway enrichment analysis (GO terms) provided by the web-based tool suite MetaboAnalyst when taking into account differentially abundant proteins (DAPs) with *p* values <0.05 or adjusted FDR *p* values <0.05. Source data are provided with this paper.

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

## Acknowledgements

This study has been funded by the Instituto de Salud Carlos III (ISCIII) through the projects CP19/00109, PI18/00550, and PI21/00074 (to F.J.O.) and co-founded by the European Union and the Fondo Europeo de Desarrollo Regional (FEDER). This research was also partially funded by the Deutsche Forschungsgemeinschaft (DFG, German Research Foundation)—416910386—GRK 2573/1 (to M.G-S.). F.J.O. (MS19/00109) is a recipient of the Miguel Servet scheme and A.L. (FI19/00045) is a recipient of a Contrato Predoctoral de Formación en Investigación en Salud (PFIS); Ministerio de Ciencia, Innovación y Universidades, Gobierno de España (ES). The authors thank the CERCA and PERIS programs (Generalitat de Catalunya) for financial and institutional support, and are indebted to the participants of the FATBANK platform, promoted by the Centro de Investigación Biomédica en Red de la Fisiopatología de la Obesidad y la Nutrición (CIBEROBN, ISCIII) and the Institut d'Investigació Biomèdica de Girona (IDIBGI) Biobank (Biobanc IDIBGI, B.0000872), integrated into the Spanish National Biobanks Network, for their collaboration and coordination. The authors are also indebted to the IMIM Microarray Analysis Service (Hospital del Mar Medical Research Institute, Barcelona, Spain) for technical guidance.

## Author contributions

All authors contributed to data interpretation and revised the manuscript. A.L., J.L., E.C-I., J.M.M-N., N.O-C., M.M.M. and F.J.O. conducted

research (hands-on conduct of the experiments and data collection). A.M-M. and I.E. participated in data computing and evaluation. M.G-S., W.S., and J.G. conducted proteomics and the corresponding quantitative analysis and data visualization. E.S., A.S-M., and W.B. prepared adipose samples from Pkp2+/− and wild-type mice and provided cardiac mesenchymal stromal cells from PKP2-dependent ACM patients and controls. W.R. and J.M.F.-R. coordinated human fat samples, clinical information, and intellectual content collection. A.L. and F.J.O. conceptualized this study, designed the research, interpreted the results, and prepared the figures. J.M.F.-R. and F.J.O. jointly supervised this work. F.J.O. wrote the paper, developed the hypothesis, secured funding, and coordinated and led the project.

## Competing interests

The authors declare no competing interests.
