## [Peer Review File · Nature Communications]

Impaired Plakophilin-2 in obesity breaks cell cycle dynamics to breed adipocyte senescenceREVIEWER COMMENTS

Reviewer #1 (Remarks to the Author):

This is a very intriguing manuscript, providing evidence of a relation between PKP2 expression and adipocyte biology. The data are novel and potentially impactful. I do have some comments/suggestions that hopefully can help advance the relevance of this study.

I do understand that the authors focus their study on adipocytes. But deficiencies in PKP2 are most impactful in terms of cardiac biology, as the authors properly note in the introduction. In fact, arrhythmogenic right ventricular cardiomyopathy is most often associated with mutations in PKP2. Insofar, the phenotype of ARVC is only cardiac. One would suspect that if PKP2 plays an important role in adipocyte biology, an effect at that level could be apparent. Yet, none has been reported. Perhaps it would be possible for these investigators to obtain adipocytes from patients with PKP2-dependent ARVC and examine their hypothesis in a system with a natural deficiency in PKP2 expression?

The vast majority of the results reported here relate to transcript levels. The correlation between mRNA and protein levels is known to be non-linear and yet, the functions ascribable to the Pkp2 gene are mostly associated with the function of the protein and not of the transcript. Evaluations of protein levels and quantification of the latter seems advisable. Perhaps proteomics analysis?

A fascinating part of this study is the apparent parallelism with recently published data where the authors examined transcriptomics and proteomics results obtained from the cardiac tissue of patients with ARVC, and then extended their analysis to animal and cellular models (Perez-Hernandez et al; PMID 35959657). In that study, the investigators concluded that PKP2 deficiency leads to activation of the DNA damage response and repair, and that DNA damage may be a fundamental mechanism of PKP2-dependent pathogenesis. It would be very interesting to see the data in the present manuscript analyzed in the context of the Perez-Hernandez et al study.

Reviewer #2 (Remarks to the Author):

The manuscript by Lluch et al. addresses the role of PLP2 in adipose tissue function and adipocyte differentiation. The authors describe a novel role for PLP2 in obesity and adipose tissue biology. The study can potentially be important for adipose tissue field and has several strengths – analysis of PLP2 expression in multiple clinical cohorts, well-performed bioinformatic analysis of published data, several in vitro adipocyte systems used.

However, the study has major weaknesses. One of those is completely lacking mechanism for PLP2 function. The study is very descriptive and even though the authors use both downregulation and overexpression of PLP2, they don't go further than analysis of gene expression. So PLP2 function in adipose tissue remains unclear.

Major points:

1) The mechanism of PLP2 function is not established and the study is only descriptive. The authors use downregulation and over-expression of PLP2 in vitro and finds several cell-cycle-regulating genes suppressed. However, it is not clear how PLP2 is functioning to downregulate these genes, it is also not clear if downregulation of cell genes has any effect on observed changes in adipogenic genes or adipocyte phenotype. PLP2 signalling cascade leading to gene expression changes is not studied (except few phosphorylation), DNA synthesis or proliferation is not measured. Therefore, causative relationship between PLP2, regulated genes and adipocyte phenotype is not established. These questions should be addressed experimentally.

2) Second major point is discrepancy in adipogenic regulation of PLP2 and the effect of it's knockdown. The upregulation of protein expression during adipogenesis is usually connected with the role of this protein in the function of mature fat cell. However, downregulation of PLP2 enhances formation of functional adipocyte. How this comes together? What is the mechanism for that? The effect on cell cycle and senescence genes does not explain this phenomena and the authors don't make an attempt to explain it. This is definitely important for understanding PLP2 function in WAT.

3) The authors show that PLP2 is equally expressed in SVF and adipocytes in scWAT and even higher in SVF in vsWAT. Having in mind that adipocytes constitute only around 30% of adipose tissue cells, all measurements of PLP2 expression in WAT don't tell much about PLP2 function in mature adipocytes. Therefore, it is unclear in which cell type PLP2 is regulated during obesity and how that relates to adipocyte phenotype. In this context, the

results obtained from clinical data are not conclusive.

Some minor points:

- Figure 1. How many women were examined for the data shown in 1c. Preadipocytes from obese donors are known to differentiate worse than preadipocytes obtained from lean individuals. Data shown in 1c can simply be connected with lower differentiation capacity of omental preadipocytes. Please show other adipogenesis regulated genes, such as HSL, PLIN, ADIPOQ.
- Please, show the kinetics of knockdown and overexpression efficiency of PLP2 by western (transient vs sustained).
- English language should be checked and edited where needed.

Reviewer #3 (Remarks to the Author):

In this research article, the Authors aimed at dissecting the role of Plakophilin-2 (PKP2) during adipogenesis in pre-adipocytes (PA), within the context of depot specific locations, including subcutaneous (SC) or omental (OM) PA. To this end, the Authors performed analyses of depot-related PKP2 gene expression in PA obtained from non-obese (NO), as well as lean or obese women, clustered on BMI bases, monitoring the PKP2 patterning in the resulting adipocytes. They investigated PKP2 expression during adipogenesis within the context of an in vitro induced pro-inflammatory microenvironment, as well as in 3T3-L1 fibroblasts during their commitment to an adipogenic lineage. The Authors investigated the consequences elicited by diminished (i.e. lentiviral induced) PKP2 expression in 3T3-L1 preadipocytes on their differentiation into lipid-containing fat cells. The pattern of PKP2 was followed in SC and OM adipocytes from obese and lean subjects, even assessing the consequences of BMI reduction owing to significant weight loss after bariatric surgery, and consequent reduction in caloric intake. The study was corroborated by wide ranging transcriptional profiling to identify the putative up or downregulated transcripts under the different experimental conditions. In summary, the Authors found that:

- PKP2 expression peaked in terminally differentiated adipocytes

- A downregulation of PKP2 occurred in adipocytes under a pro-inflammatory microenvironment.
- PKP2 had a significant role in maintaining a gene expression program required to sustain cell cycle.
- PKP2 was downregulated in SC adipose cells from obese subjects, being normalized upon mild-to-intense weight loss.
- Impaired PKP2 levels in human adipocytes (elicited by a pro-inflammatory context or artificial gene downregulation) inhibited cell cycle dynamics ensuing in premature senescence, and increased intracellular lipid accumulation.
- Restoring PKP2 in inflamed adipocytes led to cell cycle re-activation, counteracting senescence progression.

On the whole, the Authors' claim that "their findings bound the expression of PKP2 in fat cells to the physiopathology of obesity, and uncover a previously unknown defect in cell cycle and activated adipocyte senescence due to impaired PKP2" appears to be substantiated by the provided experimental evidence. The methodological approach is state-of-the-art.

Besides this, I have noticed at page 6 (lines 209-210): "and thus inversely associated with BMI and the expression of leptin (LEP) (Figure 2e)". In Figure 2, this is depicted in panel f (Figure 2f).

On the same page 6 (lines 211-2014): "In contrast with results in SC adipose tissue, measures of PKP2 in OM fat revealed no relationship with BMI (Figure 2f-g, in the text, while this is shown in panels e, g), "yet slightly decreased expressions in non-obese subjects with IGT" (Figure 2f, indeed shown in panel e).

As a minor remark, in my humble opinion the Authors should avoid using sentences that overemphasize their own results, like "seminal findings", "strikingly" ...

We thank all the Reviewers for their constructive comments and insightful suggestions. We have addressed these in full, either experimentally or with explanations. Any alterations to the original manuscript text are pasted into this rebuttal document, coloured in red, and encapsulated by quotation marks. Citations introduced into the rebuttal letter are included in a **References** section at the end of this document.

Reviewer #1 (Remarks to the Author):

This is a very intriguing manuscript, providing evidence of a relation between PKP2 expression and adipocyte biology. The data are novel and potentially impactful. I do have some comments/suggestions that hopefully can help advance the relevance of this study.

I do understand that the authors focus their study on adipocytes. But deficiencies in PKP2 are most impactful in terms of cardiac biology, as the authors properly note in the introduction. In fact, arrhythmogenic right ventricular cardiomyopathy is most often associated with mutations in PKP2. Insofar, the phenotype of ARVC is only cardiac. One would suspect that if PKP2 plays an important role in adipocyte biology, an effect at that level could be apparent. Yet, none has been reported. Perhaps it would be possible for these investigators to obtain adipocytes from patients with PKP2-dependent ARVC and examine their hypothesis in a system with a natural deficiency in PKP2 expression?

We thank the Reviewer for his/her insightful suggestion. For the current revision of our work, we were committed to address all the concerns raised by the Referees with new data. With regard to the examination of our hypothesis in fat cells from ARVC patients, Dr Elena Sommariva and co-workers were kind enough to contribute to our study by providing human adipose-derived stromal cells (**Table S5**)^{1,2}, and also samples of bulk SC adipose tissue, and SVC and adipocytes from mice with *Pkp2* haploinsufficiency (*Pkp2*^{+/-})^{3,4} were collected and analysed in the context of the current work (**Table S6**). Thereby, we were able to investigate the effects of PKP2 deficiency on adipose tissue and adipose-derived cells (**Figure 5a**), and in mesenchymal stromal cells obtained from subcutaneous fat biopsy samples in PKP2-dependent ARVC patients, both directly (dry pellet) and after being plated in culture and stimulated with adipogenic conditions (**Figure 5b**). Then, we assessed the expression of a battery of adipocyte biomarkers, while gene expression patterns suggestive of impaired cell cycle were consistently outlined along this research both in mouse (**Figure 5c**) and human adipose-derived cell samples (**Figure 5d-f**). We have included these results under the headline “*Querying adipose cells of *Pkp2*^{+/-} mice and PKP2-dependent ACM patients*” in the **Results** section of our revised manuscript as follows:

“To confirm that defective PKP2 in adipose tissue is mostly associated with gene expression patterns suggestive of impaired fat cell cycle, we took advantage of subcutaneous adipose tissue collected in a heterozygous (*Pkp2*^{+/-}) *Pkp2* knock-out mouse model^{2,53} (**Figure 5a**), and analysed subcutaneous stromal cells obtained from fat biopsy sampling in PKP2-dependent arrhythmogenic cardiomyopathy (ACM) patients (**Table S5**)^{17,54}, as illustrated in **Figure 5b**. In *Pkp2*^{+/-} mice (**Table S6**), the analysis of gene candidates related to cell cycle depicted a global trend towards decreasing expression levels in whole adipose tissue and *ex vivo* isolated adipocytes, yet no consistent effect on

adipogenesis regulated genes (*Pparg*, *Srebp1*, *Adipoq*, *Plin1*) were observed (Figure 5c). In our limited human cohort, we also did not detect significant effects on genes related to fat cell performance (*FASN*, *PLIN1*), senescence (*SQSTM1*), metabolic imbalance (*PAPPA*, *DDIT4*), or enhanced inflammation (*TNF*), but instead observed a common trend of decrease for genes belonging to cell cycle in *ex vivo* isolated *PKP2*-mutated adipose mesenchymal stromal cells, after correcting for age and sex (Figure 5d). Furthermore, when these cells were plated on plastic in preadipocytes culture medium, grown until confluence, and stimulated or not with adipogenic conditions (mature adipocytes and preadipocytes), altered expression of *CDK1*, *MKI67*, *PLK4*, and *TOP2A* was verified in *PKP2*-mutated undifferentiated and differentiated lipid-containing adipocytes (Figure 5e-f). *PAPPA*, *IMPA2*, and *MT1F* also showed expression patterns mostly linked with the genotype of *PKP2* (Figure 5e-f), regardless of potential donor-related confounders (e.g., age, sex, etc.) and the overall nature of these adipose-derived mesenchymal stromal cells when forced to differentiate into adipocytes. Altogether, although the molecular changes observed do not indicate major potential metabolic dysfunctions affecting adipose cells in ACM patients, examination of our hypothesis in alternative *ex vivo* isolated and *in vitro* cultured adipocytes and adipocyte-precursor cells confirms that *PKP2* deficiency in adipose tissue may drive the impaired expression of genes related to cell cycle, suggesting alternative pathophysiological scenarios that may contribute to *PKP2*-dependent ACM phenotype severity and warrant further investigations.”

The vast majority of the results reported here relate to transcript levels. The correlation between mRNA and protein levels is known to be non-linear and yet, the functions ascribable to the Pkp2 gene are mostly associated with the function of the protein and not of the transcript. Evaluations of protein levels and quantification of the latter seems advisable. Perhaps proteomics analysis?

We agree that this is a conceptually important validation and we have used high-resolution liquid chromatography-tandem mass spectrometry and in-depth proteome-wide analysis to examine global changes affecting human preadipocyte and adipocyte proteomes upon the knockdown of *PKP2* (Figure 4a). This yielded further insight into the shift from normal conditions in adipocytes enduring the loss of *PKP2*, and provided a *bona fide* basis to better address the specific relevance of *PKP2* in mature adipocytes and undifferentiated adipocyte progenitors (Figure 4b-i, and supplemental Table S4 and Figures S4a-f and S5a-d). Please, see below and in the **Results** section of the revised version of our manuscript, under the headline “*Impact of defective PKP2 in fat cells disclosed by untargeted proteomics*”:

“As many of the functions ascribable to *PKP2* are dependent on a sequence of structural changes at the protein level, we conducted quantitative proteomics to further substantiate our transcriptomic observations following the knockdown of *PKP2* (Figure 4a). First, we determined the proteomes of control and si-*PKP2*-targeted human adipocytes and preadipocytes using high-resolution liquid chromatography-tandem mass spectrometry. Initially, a data independent acquisition strategy and analysis using DIA-NN⁵² identified 8,214 protein groups. After dropping 52 peptides without

replication within subgroups, and upon filtering against 1,684 proteins with less than 2 peptides, 6,478 protein groups were retained for further analysis (Figure S4a-b). Hierarchical clustering (Figure S4c) and the correlation plot (Figure S4d) of results obtained in each proteome yielded clear segregation of biological replicates into their respective subgroups. Principal component analysis (PCA) also showed that adipocytes (Figure 4b) and preadipocytes (Figure 4c) challenged with si-RNAs targeting PKP2 were readily distinguished from corresponding controls. Differentially abundant proteins (DAPs) were assessed by employing Bayesian moderated t-test (p-value<0.05). Amongst these, 461 DAPs (346 up and 115 downregulated) in adipocytes (Figure 4d), and 406 (219 up and 187 downregulated) in preadipocytes (Figure 4e) were shortlisted after correcting for multiple hypothesis testing (FDR). Of note, a number of the proteins that were downregulated in adipocyte cultures with defective PKP2 were associated with the gene ontology terms *Cell cycle* (GO:0007049) and *Focal adhesion* (GO:0005925), while DAPs increased in this experimental setting pointed at the *Cell response to stress* (GO:0033554), as highlighted in Figure 4f, and the supplemental Table S4 and Figures S5a and S5b. Conversely, si-PKP2 preadipocyte proteomes were characterized by the lower abundance of proteins related to the *Response to stress* (GO:0006950), also accompanying impaired *Cell cycle* (Figure S5c) and *Focal adhesion* (Figure 4g), while being overrepresented for proteins of the *Carbohydrate derivative metabolic process* (GO:1901135), amongst others (Figure S5d and Table S4). Notably, cumulative frequency of changes affecting proteins involved in these processes (Figure 4h) highlighted the degree of coordination between differentially regulated proteins in common enriched gene ontology categories in mature adipocytes and undifferentiated precursor cells with impaired PKP2 (Figure S4e). Amongst these, the decreased presence of cell cycle-related key proteins such as E2F4, ERBB2, KI67 (*MKI67*), and TOPK (*TOP2A*), and the altered abundance of enzymes mainly responsible for the early stages of the biosynthesis of lipids (*SCD*, *ACLY*), together with enhanced PAPP1 (*PAPPA*), *IMPA2* and Insulin-like growth factor binding protein 3 (*IGFBP3*, also known as *IBP3*), and decreased Metallothionein 1F (*MT1F*) protein levels in adipocyte and preadipocyte cell cultures (Figure 4i), further validated our transcriptomic results in mature adipocytes, and confirm a transitional state of PKP2-deficient precursor cells to acquire some adipocyte features under conditions that *per se* do not trigger the adipogenic transformation (Figure S4f)."

A fascinating part of this study is the apparent parallelism with recently published data where the authors examined transcriptomics and proteomics results obtained from the cardiac tissue of patients with ARVC, and then extended their analysis to animal and cellular models (Perez-Hernandez et al; PMID 35959657). In that study, the investigators concluded that PKP2 deficiency leads to activation of the DNA damage response and repair, and that DNA damage may be a fundamental mechanism of PKP2-dependent pathogenesis. It would be very interesting to see the data in the present manuscript analysed in the context of the Perez-Hernandez et al study.

Following the Reviewer's suggestion, we have measured concentrations of hydrogen peroxide (H₂O₂) in conditioned media supernatants obtained from cell cultures of either

NTC or si-*PKP2* adipocytes, and in MCM-inflamed adipose cells containing a plasmid with an empty cassette (reference) or coding for *PKP2*. These measures were conducted twice in freshly collected supernatants from two independent experiments. **Figure 6g** shows results for one of these assays (n=8 biological replicates/condition). In line with data reported in reference ⁷, H_2O_2 levels in *PKP2*-deficient cells were consistently higher than in the media of control cells, thus depicting increased oxidant production that may, in turn, propagate deleterious consequences to neighbour cells. Intriguingly, MCM did not show the same effect, but rather reduced the amount of H_2O_2 released to the media (**Figure 6g**). Nevertheless, engineered human adipocyte cells containing the plasmid coding for *PKP2* (MCM+OE_ *PKP2*) displayed even lower amounts of H_2O_2 in the media than inflamed cells treated with the plasmid control. Clearly, whether increased oxidant production in mature adipocytes lacking *PKP2* impacts adjacent fat cells warrants additional investigations. However, we believe that additional experiments on this regard are outside the scope of the present paper. Thus, in the current revised version of our manuscript, we have added the following:

“We also evaluated changes in oxidant production, related to the loss of nuclear envelope integrity and DNA damage in *PKP2*-defective cardiac cells ⁶⁴, by assessing H_2O_2 in the media, and conducted immunofluorescence staining of the cell cycle driver Cyclin D1 (CCND1) ^{65,66} and Protein Kinase C α (PKC α), a scaffold regulator linking signal transduction pathways and cell-cycle machinery ⁶⁷ that is directly influenced by the lack of *PKP2* ¹. These additional tests confirmed significant variations in the expression of key genes previously identified in human adipocyte cell cultures with impaired *PKP2* (**Figure 6b**), mimicking to a great extent the changes resulting from an inflammatory microenvironment, also characterized by decreased *PKP2* (**Figure 6c**).”

“We further confirmed the increased presence of PAPPA at the protein level (**Figure 6f**), also accompanied by the high release of H_2O_2 into the culture media of *PKP2*-deficient adipocytes (**Figure 6g**), and found Cyclin D1 immunostaining to be significantly compromised (**Figure 6h**), while the appearance of nuclear PKC α (**Figure S6e-f**) was higher in adipocytes that had been challenged with synthetic oligonucleotides against *PKP2*, as well as in those under the influence of MCM (**Figure 6i**).”

“ H_2O_2 levels (**Figure 6g**), impaired CCND1 (**Figure 6h** and **6k**), the appearance of nuclear PKC α (**Figure 6i** and **S6e**), and deranged amounts of PAPPA (**Figure 6k**) were also modified in inflamed adipocytes under treatment.”

Reviewer #2 (Remarks to the Author):

The manuscript by Lluch et al. addresses the role of PKP2 in adipose tissue function and adipocyte differentiation. The authors describe a novel role for PKP2 in obesity and adipose tissue biology. The study can potentially be important for adipose tissue field and has several strengths – analysis of PKP2 expression in multiple clinical cohorts, well-performed bioinformatic analysis of published data, several *in vitro* adipocyte systems used.

However, the study has major weaknesses. One of those is completely lacking mechanism for PKP2 function. The study is very descriptive and even though the authors use both downregulation and overexpression of PKP2, they don't go further than analysis of gene expression. So PKP2 function in adipose tissue remains unclear.

As stated in our article, the functions (potentially) ascribed to PKP2 are multiple and far from being completely understood even in cardiomyocytes. Besides its role in cell adhesion, structural maintenance, calcium cycling, cardiac rhythm, and voltage-gated ATP-sensitive potassium and sodium channel complexes, PKP2 scaffolds a signalling hub of proteins that, when translocated to the nucleus, can modify fundamental gene transcription programs¹²⁻¹⁴. Bearing this in mind, and after confirmation of PKP2 expression dynamics in adipose tissue/adipocytes and the altered expression patterns found in obese subjects, our main objective was to identify a deranged gene expression program delineating the molecular pathogenesis of diminished PKP2 in obese/inflamed adipocytes. For this revision, we were sensitive to the observations made by the Referee and additional experiments have been conducted in order to address the potential mechanism(s) downstream impaired PKP2 function in human adipocytes.

Major points:

1) The mechanism of PKP2 function is not established and the study is only descriptive. The authors use downregulation and over-expression of PKP2 *in vitro* and finds several cell-cycle-regulating genes suppressed. However, it is not clear how PKP2 is functioning to downregulate these genes, it is also not clear if downregulation of cell genes has any effect on observed changes in adipogenic genes or adipocyte phenotype. PKP2 signalling cascade leading to gene expression changes is not studied (except few phosphorylation), DNA synthesis or proliferation is not measured. Therefore, causative relationship between PKP2, regulated genes and adipocyte phenotype is not established. These questions should be addressed experimentally.

We carried out quantitative proteomics and in-depth proteome-wide analysis to identify functional variations following the knock-down of PKP2 in human preadipocytes and adipocytes (**Figure 4a**). This yielded further insight into the shift from normal conditions in fat cells enduring the loss of this scaffold protein, and provided a *bona fide* basis to better address the specific relevance of PKP2 in fully differentiated and precursor fat cells (**Figure 4b-i**, and supplemental **Table S4** and **Figures S4a-f** and **S5a-d**). On the other hand, the causative relationship between defective PKP2 and alterations affecting cell cycle-related genes is now better endorsed by results assessed in fat samples collected in a heterozygous *Pkp2*^{+/-} mouse model (**Figure 5a-b**) and adipose mesenchymal stromal cells of PKP2-dependent arrhythmogenic cardiomyopathy (ACM) patients (**Figure 5c-f**). We also provide the impact in cell growth (**Figure 1i**) and changes affecting oxidant production (related to the loss of nuclear envelope integrity and DNA

damage in PKP2-defective cardiac cells ⁷) in our cell models with altered PKP2 levels (**Figure 6g**). Finally, we examined the hypothesis that sets nuclear Protein Kinase C α (PKC α), a scaffold regulator linking signal transduction pathways and cell-cycle machinery ^{10,15} directly affected by defective PKP2 ¹¹, at the crossroad of inflammatory events affecting PKP2 signalling and the expression of cell cycle-related genes/proteins of apparent relevance to adipocyte functioning. As the Referee can see below and in the revised version of our manuscript, such experiments confirmed significant alterations in adipocyte biology driven by impaired PKP2 and potentially related to the subcellular location of PKC α (**Figure 6i** and **Figures S6e-f**). We believe that our efforts to set up multiple representative *in vitro* models for the study of this novel factor in the obesity arena, including the inclusion of results in different cell systems and exhaustive analysis of transversal/longitudinal human patient cohorts, represent a strength, reinforcing the causal relationship between our gene candidate and the activation of pathological mechanisms in obese/inflamed adipocytes, thus narrowing the relevance of this molecule in the field.

“Initially, we observed impaired growth prior to confluence (**Figure 1i**), recapitulating multiple lines of evidence linking PKP2 to cell proliferation ²⁵⁻²⁷.”

“We also evaluated changes in oxidant production, related to the loss of nuclear envelope integrity and DNA damage in PKP2-defective cardiac cells ⁶⁴, by assessing H₂O₂ in the media, and conducted immunofluorescence staining of the cell cycle driver Cyclin D1 (CCND1) ^{65,66} and Protein Kinase C α (PKC α), a scaffold regulator linking signal transduction pathways and cell-cycle machinery ⁶⁷ that is directly influenced by the lack of PKP2 ¹.”

“We further confirmed the increased presence of PAPPA at the protein level (**Figure 6f**), also accompanied by the high release of H₂O₂ into the culture media of PKP2-deficient adipocytes (**Figure 6g**), and found Cyclin D1 immunostaining to be significantly compromised (**Figure 6h**), while the appearance of nuclear PKC α (**Figure S6e-f**) was higher in adipocytes that had been challenged with synthetic oligonucleotides against PKP2, as well as in those under the influence of MCM (**Figure 6i**).”

2) *Second major point is discrepancy in adipogenic regulation of PKP2 and the effect of its knockdown. The upregulation of protein expression during adipogenesis is usually connected with the role of this protein in the function of mature fat cell. However, downregulation of PKP2 enhances formation of functional adipocyte. How this comes together? What is the mechanism for that? The effect on cell cycle and senescence genes does not explain this phenomenon and the authors don't make an attempt to explain it. This is definitely important for understanding PKP2 function in WAT.*

We thank the Reviewer for the opportunity to explain these counter-intuitive results. During the last few decades, investigators have embarked on a systematic endeavour to assess the biology of fat precursor cells and define the nature of lipid-containing adipocytes. In within, many of us struggle to comprehend the transcriptional events directing preadipocyte differentiation (adipogenesis), which is regulated by an intricate

network of transcription factors coordinating the expression of hundreds of proteins, defining the adipocyte phenotype¹⁹. Indeed, committed preadipocytes arise from multipotent mesenchymal progenitor cells found in adipose tissues, and show expression patterns and molecular features very distant from those in which terminally differentiated fat cells rely to perform their activity. As a matter of fact, sequential activation and altered expression of over 2,500 genes engage the entire terminal differentiation process^{20,21}. In this context, evidence compiled during our research points out two scenarios: first, preadipocytes with low PKP2 signal (e.g., sh/si-PKP2-treated and SC preadipocytes) show molecular insights of an emerging adipogenic phenotype, and are more prone to the adipogenic conversion than those with high amounts of PKP2 (OM preadipocytes). This observation aligns with a number of publications depicting the adipogenic component of uncommon cardiomyopathies diagnosed in patients with mutations affecting *PKP2*^{2,12}. Accordingly, the molecular basis of these diseases relies on cardiac progenitor cells that go awry and differentiate into adipocyte (phenotype-like) cells. Apparently, the pathogenesis of adipocytes replacing cardiac myocytes involves the partial suppression of canonical Wnt signalling^{12,22}, which stands for a strong deterrent of adipogenesis²³⁻²⁵. On the other hand, PKP2 is a key component of desmosomes necessary for maintaining tight junction-associated transmembrane communication, cell adhesion and interaction between neighbour cells, which are recognised requirements for adipogenesis²⁶⁻²⁸. In this context, increased PKP2 during adipogenesis should guarantee junction dynamics and fulfil the necessity of cell-to-cell contact, allowing adipocyte differentiation, commitment and survival in the complex landscape of the adipose tissue microenvironment. Therefore, a balance between the positive and negative regulation of PKP2 might be essential to have a controlled effect on adipogenesis as well as in adipocyte function, as diagrammatically represented now in **Figure 7**. Our results indicate that this balance is disrupted in obese subjects, thus compromising adipocyte wellbeing and ultimate viability. This leads to the second scenario, in which inflammatory events occurring in adipose tissue may drive defective PKP2 renewal, impaired desmosomal plaque assembly, hampered cell-to-cell contact, altered gene expression patterns, compromised cell function, and ensuing adipocyte senescence. Finally, it should be noted that there are some other examples of proteins upregulated during adipogenesis (and thus bounded to the phenotype of mature fat cells) whose downregulation enhances formation of functional adipocytes from precursor cells. For instance, the canonical Wnt/ β -catenin pathway, a well-recognised mitogen-driven cell cycle regulator and potent inhibitor of adipogenesis, plays in mature fat cells important roles in maintaining the adipocyte phenotype^{30,31}. Also, while PKC α levels are transiently elevated during the differentiation of 3T3-L1 and 3T3-F442A adipocytes, both the knockdown and chemical inhibition of this isoform (amongst others) can facilitate adipogenesis, suggesting the negative impact of this kinase in the development of new adipocytes (reviewed in²⁹). Our compiled results show consistent variations affecting the appearance of nuclear PKC α in adipocyte cultures in which PKP2 levels were modulated. These observations are consistent with data from different models whereby the interaction between PKP2 and PKC isoenzymes, either directly or in conjunction with other scaffolding proteins, has been reported^{11,32}. In these

precedents, the notion that intracellular translocation of activated PKC α may regulate molecular events linking signal transduction pathways to cell-cycle machinery is also sustained by compelling evidence readily available in the literature ^{33,34}. Thus, the underlying mechanism in adipocyte-specific PKP2 deficiency may comprise the translocation of nuclear PKC α , and supports the hypothesis that PKP2 prevents undesirable molecular patterns in adipocytes by locally tying PKC α to the cytoplasm. By doing so, PKP2 may avoid changes in the distribution of active PKC α and the phosphorylation of PKC nuclear substrates related to the synthesis of proteins involved in cytoskeletal dynamics, junction assembly, cell cycle, and adipocyte differentiation and maintenance. See now in the **Discussion** section of this revised manuscript:

“In these cell models, adipocytes presented attributes related to cellular senescence, including decreased amounts of the cell cycle driver cyclin D1, enhanced β -galactosidase activity and the subsequent release of H₂O₂, as well as changes in gene expression and protein biomarkers suggestive of impaired cell cycle, altered abundance of junctional proteins, and activated senescence, while the exogenous plasmid-based expression of human PKP2 seems to be sufficient to mitigate the chain of events leading to adipocyte failure. At the crossroad of these events able to further the molecular pathophysiology of obesity, we found consistent variations affecting the appearance of nuclear Protein Kinase C alpha (PKC α) in adipocyte cultures in which PKP2 levels were modulated. These results are consistent with data from different models whereby the interaction between PKP2 and PKC isoenzymes, either directly or in conjunction with other scaffolding proteins, was reported ^{1,86}. It is well recognised that calcium-activated, phospholipid-dependent PKC activity is involved in cellular functions ranging from growth control to intercellular adhesion, activation of cellular function, differentiation, and cell death ^{87,88}. The notion that intracellular translocation of activated PKC α regulates molecular events linking signal transduction pathways to cell-cycle machinery is also supported by compelling evidence readily available in the literature ^{89,90}. The underlying mechanism in adipocyte-specific PKP2 deficiency potentially comprises the translocation of nuclear PKC α , and supports the hypothesis that PKP2 prevents aberrant molecular patterns in adipocytes by locally tying PKC α to the cytoplasm, thus preventing changes in the distribution of active PKC α and the phosphorylation of PKC nuclear substrates related to the synthesis of proteins involved in cytoskeletal dynamics, junction assembly, cell cycle, and adipocyte differentiation and maintenance.”

3) *The authors show that PKP2 is equally expressed in SVF and adipocytes in scWAT and even higher in SVF in vsWAT. Having in mind that adipocytes constitute only around 30% of adipose tissue cells, all measurements of PKP2 expression in WAT don't tell much about PKP2 function in mature adipocytes. Therefore, it is unclear in which cell type PKP2 is regulated during obesity and how that relates to adipocyte phenotype. In this context, the results obtained from clinical data are not conclusive.*

We understand the scepticism of the Referee with regard to the significance of changes affecting PKP2 gene expression in bulk adipose tissue. Indeed, we must be very cautious when interpreting whole-tissue datasets, as major variations could arise from changes

in cell populations and/or molecular events affecting specific cell types. Certainly, results provided for SC PKP2 seem difficult to reconcile with the data acquired in OM adipose tissue. Yet, an explanation is plausible. Regarding the measurements of *PKP2* in isolated SVC and adipocytes, it should be noted that our own results and those of Vijay *et al.*³⁵ were obtained in morbid obese patients. In these subjects, adipocyte-specific *PKP2* is likely to be compromised, thus the lack of differences between SC/OM-derived adipocytes and SC SVC. On the other hand, high expression of *PKP2* in OM SVC mostly acknowledges the abundance of mesothelial cells within OM adipose tissue, while being almost absent in SC fat^{36,37}. The paradigm is nicely endorsed by Norreen-Thorsen *et al.* in reference³⁸. Such datasets served to illustrate in our revised supplemental **Figure S2e** the apparent overrepresentation of *PKP2* in SC adipocytes, while in OM adipose tissue this armadillo-repeat protein is prevalent in mesothelial cells. However, only with the analysis of extended population-based fat cell samples this relationship can be validated further. Unfortunately, isolated fat cells from extended human cohorts are not available at the moment, which compromises our ability to address the question as we would have liked to do. However, we are sensitive to the suggestion made by the Referee and this limitation has been noted in the **Results** section of our revised article (see below).

“Consistently, measures of *PKP2* gene expression taken in *ex vivo* isolated mature adipocytes (MA) and the stromal vascular cell (SVC) fraction of morbid obese patients were suggestive of different cell populations growing within OM (but not SC) SVC, as shown by our RT-PCR results (**Figure 2i**) and the single-cell RNA sequencing of Vijay *et al.*³⁵ (**Figure S2d**). This matches the enrichment of *PKP2* mRNA in SC adipocytes, while being more likely associated with the expression of biomarkers of mesothelial cells within OM fat (**Figure S2e**)³⁶. However, because these observations in bulk adipose tissue and adipose-derived cell samples were not population-based, we cannot exclude that the apparent relationship with obesity (results in SC) and other clinical characteristics (OM and SC) relies on the abundance of specific fat cell subgroups regulated during obesity. Nevertheless, our experiments *in vitro*, together with the systematic scrutiny of multiple human datasets and observations made in obese subjects following weight loss, support to a large extent the connexion between low *PKP2* in SC adipose tissue/ adipocytes and the burden of obesity/ inflammation. The exact mechanism whereby sex, age, and/or the metabolic status may influence the expression of *PKP2* in the even more complex cellularity of OM adipose tissue is a key question we seek to answer in our future studies.”

Some minor points:

- Figure 1. How many women were examined for the data shown in 1c. Preadipocytes from obese donors are known to differentiate worse than preadipocytes obtained from lean individuals. Data shown in 1c can simply be connected with lower differentiation capacity of omental preadipocytes. Please show other adipogenesis regulated genes, such as HSL, PLIN, ADIPOQ.

We thank the Referee for his/her kind suggestion. We have added plots depicting expression dynamics for well-recognised adipogenesis regulated genes (i.e., *ADIPOQ*, *PLIN*, and *FASN*). In **Figure 1c**, we show expression patterns for differentiating SC preadipocytes obtained from one lean (BMI<25 kg/m²) and one obese (BMI>30 kg/m²) women of approximately same age (35-40 years). We now specify it in the text. We are aware of the point made by the Referee concerning the important caveat that may represent inter-individual differences between fat precursor cells from different donors, and we cannot rule out the potential impact of other confounders, besides sex, age and weight. Only with the analysis of extended population-based cell cultures, these expression dynamics for PKP2 in differentiating human adipocyte precursor cells can be validated further. Notwithstanding this, through these pioneering assays in primary adipocyte progenitors, we wanted to test whether PKP2 was related to the adipogenic program, and how. Thereby, data shown in **Figure 1b** and **1c** indicate the unprecedented connection of PKP2 with canonical adipogenesis, and thus, attenuated differentiation capacity of obese and omental preadipocytes further substantiated this association.

- *Please, show the kinetics of knockdown and overexpression efficiency of PKP2 by western (transient vs sustained).*

The kinetics of PKP2 knockdown and overexpression efficiency at protein levels are shown in **Figures 3b, 6i** and **6h**.

- *English language should be checked and edited where needed.*

English language has been edited along the manuscript, and a few stylistic changes have been made to this revised version.

Reviewer #3 (Remarks to the Author):

In this research article, the Authors aimed at dissecting the role of Plakophilin-2 (PKP2) during adipogenesis in pre-adipocytes (PA), within the context of depot specific locations, including subcutaneous (SC) or omental (OM) PA. To this end, the Authors performed analyses of depot-related PKP2 gene expression in PA obtained from non-obese (NO), as well as lean or obese women, clustered on BMI bases, monitoring the PKP2 patterning in the resulting adipocytes. They investigated PKP2 expression during adipogenesis within the context of an in vitro induced pro-inflammatory microenvironment, as well as in 3T3-L1 fibroblasts during their commitment to an adipogenic lineage. The Authors investigated the consequences elicited by diminished (i.e. lentiviral induced) PKP2 expression in 3T3-L1 preadipocytes on their differentiation into lipid-containing fat cells. The pattern of PKP2 was followed in SC and OM adipocytes from obese and lean subjects, even assessing the consequences of BMI reduction owing to significant weight loss after bariatric surgery, and consequent reduction in caloric intake. The study was corroborated by wide ranging transcriptional profiling to identify the putative up or downregulated transcripts under the different experimental conditions. In summary, the Authors found that:

- PKP2 expression peaked in terminally differentiated adipocytes*
- A downregulation of PKP2 occurred in adipocytes under a pro-inflammatory microenvironment.*
- PKP2 had a significant role in maintaining a gene expression program required to sustain cell cycle.*
- PKP2 was downregulated in SC adipose cells from obese subjects, being normalized upon mild-to-intense weight loss.*
- Impaired PKP2 levels in human adipocytes (elicited by a pro-inflammatory context or artificial gene downregulation) inhibited cell cycle dynamics ensuing in premature senescence, and increased intracellular lipid accumulation.*
- Restoring PKP2 in inflamed adipocytes led to cell cycle re-activation, counteracting senescence progression.*

On the whole, the Authors' claim that "their findings bound the expression of PKP2 in fat cells to the physiopathology of obesity, and uncover a previously unknown defect in cell cycle and activated adipocyte senescence due to impaired PKP2" appears to be substantiated by the provided experimental evidence. The methodological approach is state-of-the-art.

Besides this, I have noticed at page 6 (lines 209-210): "and thus inversely associated with BMI and the expression of leptin (LEP) (Figure 2e)". In Figure 2, this is depicted in panel f (Figure 2f).

We apologize for the lack of clarity in the previous version of this manuscript and we thank the Reviewer for his/her kindly help and suggestions. We have now taken extensive care in data labelling and description of results. See now along the revised manuscript.

On the same page 6 (lines 211-214): "In contrast with results in SC adipose tissue, measures of PKP2 in OM fat revealed no relationship with BMI (Figure 2f-g, in the text, while this is shown in panels e, g), "yet slightly decreased expressions in non-obese subjects with IGT" (Figure 2f, indeed shown in panel e).

We have amended the mistake, and hope that our new data and the revised description of available results make this section of the manuscript much clearer.

As a minor remark, in my humble opinion the Authors should avoid using sentences that overemphasize their own results, like “seminal findings”, “strikingly”,...

We have balanced the tone of our narrative along this revised version of the manuscript.

REFERENCES

1. Pilato, C. A. *et al.* Isolation and Characterization of Cardiac Mesenchymal Stromal Cells from Endomyocardial Bioptic Samples of Arrhythmogenic Cardiomyopathy Patients. *J. Vis. Exp.* **132**, 57263 (2018).
2. Sommariva, E. *et al.* Cardiac mesenchymal stromal cells are a source of adipocytes in arrhythmogenic cardiomyopathy. *Eur. Heart J.* **37**, 1835–1846 (2016).
3. Grossmann, K. S. *et al.* Requirement of plakophilin 2 for heart morphogenesis and cardiac junction formation. *J. Cell Biol.* **167**, 149–160 (2004).
4. Sommariva, E. *et al.* Oxidized LDL-dependent pathway as new pathogenic trigger in arrhythmogenic cardiomyopathy. *EMBO Mol. Med.* **13**, e14365 (2021).
5. Demichev, V., Messner, C. B., Vernardis, S. I., Lilley, K. S. & Ralser, M. DIA-NN: neural networks and interference correction enable deep proteome coverage in high throughput. *Nat. Methods* **17**, 41–44 (2020).
6. García-Marqués, F. *et al.* A Novel Systems-Biology Algorithm for the Analysis of Coordinated Protein Responses Using Quantitative Proteomics. *Mol. Cell. Proteomics* **15**, 1740–1760 (2016).
7. Pérez-Hernández, M. *et al.* Loss of Nuclear Envelope Integrity and Increased Oxidant Production Cause DNA Damage in Adult Hearts Deficient in PKP2: A Molecular Substrate of ARVC. *Circulation* **146**, 851–867 (2022).
8. Musgrove, E. A., Lee, C. S., Buckley, M. F. & Sutherland, R. L. Cyclin D1 induction in breast cancer cells shortens G1 and is sufficient for cells arrested in G1 to complete the cell cycle. *Proc. Natl. Acad. Sci. U. S. A.* **91**, 8022–8026 (1994).
9. Guardavaccaro, D. *et al.* Arrest of G(1)-S progression by the p53-inducible gene PC3 is Rb dependent and relies on the inhibition of cyclin D1 transcription. *Mol. Cell. Biol.* **20**, 1797–1815 (2000).
10. Marini, N. J. *et al.* A pathway in the yeast cell division cycle linking protein kinase C (Pkc1) to activation of Cdc28 at START. *EMBO J.* **15**, 3040–3052 (1996).
11. Bass-Zubek, A. E. *et al.* Plakophilin 2: A critical scaffold for PKC α that regulates intercellular junction assembly. *J. Cell Biol.* **181**, 605–613 (2008).
12. Chen, S. N. *et al.* The hippo pathway is activated and is a causal mechanism for adipogenesis in arrhythmogenic cardiomyopathy. *Circ. Res.* **114**, 454–468 (2014).
13. Cerrone, M. *et al.* Plakophilin-2 is required for transcription of genes that control calcium cycling and cardiac rhythm. *Nat. Commun.* **8**, 106 (2017).
14. Mertens, C. *et al.* Nuclear particles containing RNA polymerase III complexes associated with the junctional plaque protein plakophilin 2. *Proc. Natl. Acad. Sci. U. S. A.* **98**, 7795–7800 (2001).
15. Livneh, E. & Fishman, D. D. Linking protein kinase C to cell-cycle control. *Eur. J. Biochem.* **248**, 1–9 (1997).
16. Arimoto, K. *et al.* Plakophilin-2 promotes tumor development by enhancing ligand-dependent and -independent epidermal growth factor receptor dimerization and activation. *Mol. Cell. Biol.* **34**, 3843–3854 (2014).
17. Wu, Y., Liu, L., Shen, X., Liu, W. & Ma, R. Plakophilin-2 Promotes Lung Adenocarcinoma Development via Enhancing Focal Adhesion and Epithelial-Mesenchymal Transition. *Cancer Manag. Res.* **13**, 559–570 (2021).
18. Hao, X.-L. *et al.* Plakophilin-2 accelerates cell proliferation and migration through activating EGFR signaling in lung adenocarcinoma. *Pathol. Res. Pract.* **215**, 152438 (2019).
19. Cristancho, A. G. & Lazar, M. A. Forming functional fat: a growing understanding of adipocyte differentiation. *Nat. Rev. Mol. Cell Biol.* **12**, 722–734 (2011).

20. Farmer, S. R. Transcriptional control of adipocyte formation. *Cell Metab.* **4**, 263–273 (2006).
21. Rosen, E. D. & MacDougald, O. A. Adipocyte differentiation from the inside out. *Nat. Rev. Mol. Cell Biol.* **7**, 885–896 (2006).
22. Garcia-Gras, E. *et al.* Suppression of canonical Wnt/ β -catenin signaling by nuclear plakoglobin recapitulates phenotype of arrhythmogenic right ventricular cardiomyopathy. *J. Clin. Invest.* **116**, 2012–2021 (2006).
23. Christodoulides, C., Lagathu, C., Sethi, J. K. & Vidal-Puig, A. Adipogenesis and WNT signalling. *Trends Endocrinol. Metab.* **20**, 16–24 (2009).
24. Ross, S. E. *et al.* Inhibition of adipogenesis by Wnt signaling. *Science* **289**, 950–953 (2000).
25. de Winter, T. J. J. & Nusse, R. Running Against the Wnt: How Wnt/ β -Catenin Suppresses Adipogenesis. *Front. cell Dev. Biol.* **9**, 627429 (2021).
26. Chen, Q. *et al.* Fate decision of mesenchymal stem cells: adipocytes or osteoblasts? *Cell Death Differ.* **23**, 1128–1139 (2016).
27. Schiller, P. C., D'Ippolito, G., Brambilla, R., Roos, B. A. & Howard, G. A. Inhibition of gap-junctional communication induces the trans-differentiation of osteoblasts to an adipocytic phenotype in vitro. *J. Biol. Chem.* **276**, 14133–14138 (2001).
28. Murakami, K. *et al.* Antiobesity Action of ACAM by Modulating the Dynamics of Cell Adhesion and Actin Polymerization in Adipocytes. *Diabetes* **65**, 1255–1267 (2016).
29. Schmitz-Peiffer, C. The tail wagging the dog--regulation of lipid metabolism by protein kinase C. *FEBS J.* **280**, 5371–5383 (2013).
30. Bagchi, D. P. *et al.* Wnt/ β -catenin signaling regulates adipose tissue lipogenesis and adipocyte-specific loss is rigorously defended by neighboring stromal-vascular cells. *Mol. Metab.* **42**, 101078 (2020).
31. Chen, M. *et al.* CTNNB1/ β -catenin dysfunction contributes to adiposity by regulating the cross-talk of mature adipocytes and preadipocytes. *Sci. Adv.* **6**, eaax9605 (2020).
32. Nagler, S. *et al.* Plakophilin 2 regulates intestinal barrier function by modulating protein kinase C activity in vitro. *Tissue barriers* 2138061 (2022).
33. Chalkiadaki, G. *et al.* Low molecular weight heparin inhibits melanoma cell adhesion and migration through a PKCa/JNK signaling pathway inducing actin cytoskeleton changes. *Cancer Lett.* **312**, 235–244 (2011).
34. Martelli, A. M., Sang, N., Borgatti, P., Capitani, S. & Neri, L. M. Multiple biological responses activated by nuclear protein kinase C. *J. Cell. Biochem.* **74**, 499–521 (1999).
35. Vijay, J. *et al.* Single-cell analysis of human adipose tissue identifies depot and disease specific cell types. *Nat. Metab.* **2**, 97–109 (2020).
36. Chau, Y.-Y. *et al.* Visceral and subcutaneous fat have different origins and evidence supports a mesothelial source. *Nat. Cell Biol.* **16**, 367–375 (2014).
37. Emont, M. P. *et al.* A single-cell atlas of human and mouse white adipose tissue. *Nature* **603**, 926–933 (2022).
38. Norreen-Thorsen, M. *et al.* A human adipose tissue cell-type transcriptome atlas. *Cell Rep.* **40**, 111046 (2022).

REVIEWERS' COMMENTS

Reviewer #1 (Remarks to the Author):

The authors have satisfactorily addressed my concerns.

Reviewer #2 (Remarks to the Author):

The authors have made an extensive revision adding important information and additional experiments requested by this reviewer.

Although some conclusions are still quite weak, the authors are open about possible pitfalls of the study in this new version. This reviewer has no additional comments.

Reviewer #3 (Remarks to the Author):

I confirm my previous analysis of this manuscript, in considering that the Authors' main claim "that their observations relate PKP2 expression in fat cells to the physiopathology of obesity, highlighting a previously unknown defect in cell cycle and activated adipocyte senescence due to impaired PKP2" appears to be substantiated by the provided experimental evidence, and by the methodological approach.

The Authors have satisfactorily amended the revised manuscript according to my previous remarks, by correcting some panel mislabeling in Figure 2, and blunting overemphasizing of their own results.

I have also noticed that the quality of the revised manuscript has further improved, following the revisions made in accordance with the suggestions and criticisms raised by the other Reviewers.

We thank all the Reviewers for their constructive comments and insightful suggestions. We have addressed these in full, either experimentally or with explanations. Any alterations to the original manuscript text are pasted into this rebuttal document, coloured in red, and encapsulated by quotation marks. Citations introduced into the rebuttal letter are included in a **References** section at the end of this document.

Manuscript Number: NCOMMS-22-35714A – 2nd version

Reviewer #1 (Remarks to the Author):

The authors have satisfactorily addressed my concerns.

We thank the Reviewer for his/her contribution to strengthening the manuscript and positive evaluation of the revised manuscript.

Reviewer #1 (Remarks to the Author):

The authors have made an extensive revision adding important information and additional experiments requested by this reviewer. Although some conclusions are still quite weak, the authors are open about possible pitfalls of the study in this new version. This reviewer has no additional comments.

We thank the Reviewer for his/her contribution to strengthening the lines of evidence presented and believe that his/her suggestion on explicitly presenting potential shortcomings of the study design has rendered it more rigorous.

Reviewer #1 (Remarks to the Author):

I confirm my previous analysis of this manuscript, in considering that the Authors' main claim "that their observations relate PKP2 expression in fat cells to the physiopathology of obesity, highlighting a previously unknown defect in cell cycle and activated adipocyte senescence due to impaired PKP2" appears to be substantiated by the provided experimental evidence, and by the methodological approach. The Authors have satisfactorily amended the revised manuscript according to my previous remarks, by correcting some panel mislabelling in Figure 2, and blunting overemphasizing of their own results. I have also noticed that the quality of the revised manuscript has further improved, following the revisions made in accordance with the suggestions and criticisms raised by the other Reviewers.

We thank the Reviewer for his/her kindly help and positive evaluation of our work.

Manuscript Number: NCOMMS-22-35714-T – 1st version

Reviewer #1 (Remarks to the Author):

This is a very intriguing manuscript, providing evidence of a relation between PKP2 expression and adipocyte biology. The data are novel and potentially impactful. I do have some comments/suggestions that hopefully can help advance the relevance of this study.

I do understand that the authors focus their study on adipocytes. But deficiencies in PKP2 are most impactful in terms of cardiac biology, as the authors properly note in the introduction. In fact, arrhythmogenic right ventricular cardiomyopathy is most often associated with mutations in PKP2. Insofar, the phenotype of ARVC is only cardiac. One would suspect that if PKP2 plays

an important role in adipocyte biology, an effect at that level could be apparent. Yet, none has been reported. Perhaps it would be possible for these investigators to obtain adipocytes from patients with PKP2-dependent ARVC and examine their hypothesis in a system with a natural deficiency in PKP2 expression?

We thank the Reviewer for his/her insightful suggestion. For the current revision of our work, we were committed to address all the concerns raised by the Referees with new data. With regard to the examination of our hypothesis in fat cells from ARVC patients, Dr Elena Sommariva and co-workers were kind enough to contribute to our study by providing human adipose-derived stromal cells (**Table S5**)^{1,2}, and also samples of bulk SC adipose tissue, and SVC and adipocytes from mice with *Pkp2* haploinsufficiency (*Pkp2*^{+/-})^{3,4} were collected and analysed in the context of the current work (**Table S6**). Thereby, we were able to investigate the effects of PKP2 deficiency on adipose tissue and adipose-derived cells (**Figure 5a**), and in mesenchymal stromal cells obtained from subcutaneous fat biopsy samples in PKP2-dependent ARVC patients, both directly (dry pellet) and after being plated in culture and stimulated with adipogenic conditions (**Figure 5b**). Then, we assessed the expression of a battery of adipocyte biomarkers, while gene expression patterns suggestive of impaired cell cycle were consistently outlined along this research both in mouse (**Figure 5c**) and human adipose-derived cell samples (**Figure 5d-f**). We have included these results under the headline “*Querying adipose cells of Pkp2*^{+/-} mice and PKP2-dependent ACM patients” in the **Results** section of our revised manuscript as follows:

“To confirm that defective PKP2 in adipose tissue is mostly associated with gene expression patterns suggestive of impaired fat cell cycle, we took advantage of subcutaneous adipose tissue collected in a heterozygous (*Pkp2*^{+/-}) *Pkp2* knock-out mouse model^{2,53} (**Figure 5a**), and analysed subcutaneous stromal cells obtained from fat biopsy sampling in PKP2-dependent arrhythmogenic cardiomyopathy (ACM) patients (**Table S5**)^{17,54}, as illustrated in **Figure 5b**. In *Pkp2*^{+/-} mice (**Table S6**), the analysis of gene candidates related to cell cycle depicted a global trend towards decreasing expression levels in whole adipose tissue and *ex vivo* isolated adipocytes, yet no consistent effect on adipogenesis regulated genes (*Pparg*, *Srebp1*, *Adipoq*, *Plin1*) were observed (**Figure 5c**). In our limited human cohort, we also did not detect significant effects on genes related to fat cell performance (*FASN*, *PLIN1*), senescence (*SQSTM1*), metabolic imbalance (*PAPPA*, *DDIT4*), or enhanced inflammation (*TNF*), but instead observed a common trend of decrease for genes belonging to cell cycle in *ex vivo* isolated PKP2-mutated adipose mesenchymal stromal cells, after correcting for age and sex (**Figure 5d**). Furthermore, when these cells were plated on plastic in preadipocytes culture medium, grown until confluence, and stimulated or not with adipogenic conditions (mature adipocytes and preadipocytes), altered expression of *CDK1*, *MKI67*, *PLK4*, and *TOP2A* was verified in PKP2-mutated undifferentiated and differentiated lipid-containing adipocytes (**Figure 5e-f**). *PAPPA*, *IMPA2*, and *MT1F* also showed expression patterns mostly linked with the genotype of PKP2 (**Figure 5e-f**), regardless of potential donor-related confounders (e.g., age, sex, etc.) and the overall nature of these adipose-derived mesenchymal stromal cells when forced to differentiate into adipocytes. Altogether,

although the molecular changes observed do not indicate major potential metabolic dysfunctions affecting adipose cells in ACM patients, examination of our hypothesis in alternative *ex vivo* isolated and *in vitro* cultured adipocytes and adipocyte-precursor cells confirms that PKP2 deficiency in adipose tissue may drive the impaired expression of genes related to cell cycle, suggesting alternative pathophysiological scenarios that may contribute to PKP2-dependent ACM phenotype severity and warrant further investigations.”

The vast majority of the results reported here relate to transcript levels. The correlation between mRNA and protein levels is known to be non-linear and yet, the functions ascribable to the Pkp2 gene are mostly associated with the function of the protein and not of the transcript. Evaluations of protein levels and quantification of the latter seems advisable. Perhaps proteomics analysis?

We agree that this is a conceptually important validation and we have used high-resolution liquid chromatography-tandem mass spectrometry and in-depth proteome-wide analysis to examine global changes affecting human preadipocyte and adipocyte proteomes upon the knockdown of PKP2 (Figure 4a). This yielded further insight into the shift from normal conditions in adipocytes enduring the loss of PKP2, and provided a *bona fide* basis to better address the specific relevance of PKP2 in mature adipocytes and undifferentiated adipocyte progenitors (Figure 4b-i, and supplemental Table S4 and Figures S4a-f and S5a-d). Please, see below and in the Results section of the revised version of our manuscript, under the headline “Impact of defective PKP2 in fat cells disclosed by untargeted proteomics”:

“As many of the functions ascribable to PKP2 are dependent on a sequence of structural changes at the protein level, we conducted quantitative proteomics to further substantiate our transcriptomic observations following the knockdown of PKP2 (Figure 4a). First, we determined the proteomes of control and si-PKP2-targeted human adipocytes and preadipocytes using high-resolution liquid chromatography-tandem mass spectrometry. Initially, a data independent acquisition strategy and analysis using DIA-NN⁵² identified 8,214 protein groups. After dropping 52 peptides without replication within subgroups, and upon filtering against 1,684 proteins with less than 2 peptides, 6,478 protein groups were retained for further analysis (Figure S4a-b). Hierarchical clustering (Figure S4c) and the correlation plot (Figure S4d) of results obtained in each proteome yielded clear segregation of biological replicates into their respective subgroups. Principal component analysis (PCA) also showed that adipocytes (Figure 4b) and preadipocytes (Figure 4c) challenged with si-RNAs targeting PKP2 were readily distinguished from corresponding controls. Differentially abundant proteins (DAPs) were assessed by employing Bayesian moderated t-test (p-value<0.05). Amongst these, 461 DAPs (346 up and 115 downregulated) in adipocytes (Figure 4d), and 406 (219 up and 187 downregulated) in preadipocytes (Figure 4e) were shortlisted after correcting for multiple hypothesis testing (FDR). Of note, a number of the proteins that were downregulated in adipocyte cultures with defective PKP2 were associated with the gene ontology terms *Cell cycle* (GO:0007049) and *Focal adhesion* (GO:0005925), while DAPs increased in this experimental setting pointed at the *Cell response to stress*

(GO:0033554), as highlighted in **Figure 4f**, and the supplemental **Table S4** and **Figures S5a** and **S5b**. Conversely, si-PKP2 preadipocyte proteomes were characterized by the lower abundance of proteins related to the *Response to stress* (GO:0006950), also accompanying impaired *Cell cycle* (**Figure S5c**) and *Focal adhesion* (**Figure 4g**), while being overrepresented for proteins of the *Carbohydrate derivative metabolic process* (GO:1901135), amongst others (**Figure S5d** and **Table S4**). Notably, cumulative frequency of changes affecting proteins involved in these processes (**Figure 4h**) highlighted the degree of coordination between differentially regulated proteins in common enriched gene ontology categories in mature adipocytes and undifferentiated precursor cells with impaired PKP2 (**Figure S4e**). Amongst these, the decreased presence of cell cycle-related key proteins such as E2F4, ERBB2, KI67 (*MKI67*), and TOPK (*TOP2A*), and the altered abundance of enzymes mainly responsible for the early stages of the biosynthesis of lipids (SCD, ACLY), together with enhanced PAPP1 (*PAPPA*), IMPA2 and Insulin-like growth factor binding protein 3 (*IGFBP3*, also known as *IBP3*), and decreased Metallothionein 1F (*MT1F*) protein levels in adipocyte and preadipocyte cell cultures (**Figure 4i**), further validated our transcriptomic results in mature adipocytes, and confirm a transitional state of PKP2-deficient precursor cells to acquire some adipocyte features under conditions that *per se* do not trigger the adipogenic transformation (**Figure S4f**)."

A fascinating part of this study is the apparent parallelism with recently published data where the authors examined transcriptomics and proteomics results obtained from the cardiac tissue of patients with ARVC, and then extended their analysis to animal and cellular models (Perez-Hernandez et al; PMID 35959657). In that study, the investigators concluded that PKP2 deficiency leads to activation of the DNA damage response and repair, and that DNA damage may be a fundamental mechanism of PKP2-dependent pathogenesis. It would be very interesting to see the data in the present manuscript analysed in the context of the Perez-Hernandez et al study.

Following the Reviewer's suggestion, we have measured concentrations of hydrogen peroxide (H₂O₂) in conditioned media supernatants obtained from cell cultures of either NTC or si-PKP2 adipocytes, and in MCM-inflamed adipose cells containing a plasmid with an empty cassette (reference) or coding for *PKP2*. These measures were conducted twice in freshly collected supernatants from two independent experiments. **Figure 6g** shows results for one of these assays (n=8 biological replicates/condition). In line with data reported in reference ⁷, H₂O₂ levels in *PKP2*-deficient cells were consistently higher than in the media of control cells, thus depicting increased oxidant production that may, in turn, propagate deleterious consequences to neighbour cells. Intriguingly, MCM did not show the same effect, but rather reduced the amount of H₂O₂ released to the media (**Figure 6g**). Nevertheless, engineered human adipocyte cells containing the plasmid coding for *PKP2* (MCM+OE_ *PKP2*) displayed even lower amounts of H₂O₂ in the media than inflamed cells treated with the plasmid control. Clearly, whether increased oxidant production in mature adipocytes lacking *PKP2* impacts adjacent fat cells warrants additional investigations. However, we believe that additional experiments on this

regard are outside the scope of the present paper. Thus, in the current revised version of our manuscript, we have added the following:

“We also evaluated changes in oxidant production, related to the loss of nuclear envelope integrity and DNA damage in *PKP2*-defective cardiac cells ⁶⁴, by assessing H_2O_2 in the media, and conducted immunofluorescence staining of the cell cycle driver Cyclin D1 (*CCND1*) ^{65,66} and Protein Kinase $C\alpha$ (*PKC\alpha*), a scaffold regulator linking signal transduction pathways and cell-cycle machinery ⁶⁷ that is directly influenced by the lack of *PKP2* ¹. These additional tests confirmed significant variations in the expression of key genes previously identified in human adipocyte cell cultures with impaired *PKP2* (**Figure 6b**), mimicking to a great extent the changes resulting from an inflammatory microenvironment, also characterized by decreased *PKP2* (**Figure 6c**).”

“We further confirmed the increased presence of PAPPA at the protein level (**Figure 6f**), also accompanied by the high release of H_2O_2 into the culture media of *PKP2*-deficient adipocytes (**Figure 6g**), and found Cyclin D1 immunostaining to be significantly compromised (**Figure 6h**), while the appearance of nuclear *PKC\alpha* (**Figure S6e-f**) was higher in adipocytes that had been challenged with synthetic oligonucleotides against *PKP2*, as well as in those under the influence of MCM (**Figure 6i**).”

“ H_2O_2 levels (**Figure 6g**), impaired *CCND1* (**Figure 6h** and **6k**), the appearance of nuclear *PKC\alpha* (**Figure 6i** and **S6e**), and deranged amounts of PAPPA (**Figure 6k**) were also modified in inflamed adipocytes under treatment.”

Reviewer #2 (Remarks to the Author):

The manuscript by Lluch et al. addresses the role of PKP2 in adipose tissue function and adipocyte differentiation. The authors describe a novel role for PKP2 in obesity and adipose tissue biology. The study can potentially be important for adipose tissue field and has several strengths – analysis of PKP2 expression in multiple clinical cohorts, well-performed bioinformatic analysis of published data, several in vitro adipocyte systems used.

However, the study has major weaknesses. One of those is completely lacking mechanism for PKP2 function. The study is very descriptive and even though the authors use both downregulation and overexpression of PKP2, they don't go further than analysis of gene expression. So PKP2 function in adipose tissue remains unclear.

As stated in our article, the functions (potentially) ascribed to PKP2 are multiple and far from being completely understood even in cardiomyocytes. Besides its role in cell adhesion, structural maintenance, calcium cycling, cardiac rhythm, and voltage-gated ATP-sensitive potassium and sodium channel complexes, PKP2 scaffolds a signalling hub of proteins that, when translocated to the nucleus, can modify fundamental gene transcription programs¹²⁻¹⁴. Bearing this in mind, and after confirmation of PKP2 expression dynamics in adipose tissue/adipocytes and the altered expression patterns found in obese subjects, our main objective was to identify a deranged gene expression program delineating the molecular pathogenesis of diminished PKP2 in obese/inflamed adipocytes. For this revision, we were sensitive to the observations made by the Referee and additional experiments have been conducted in order to address the potential mechanism(s) downstream impaired PKP2 function in human adipocytes.

Major points:

1) *The mechanism of PKP2 function is not established and the study is only descriptive. The authors use downregulation and over-expression of PKP2 in vitro and finds several cell-cycle-regulating genes suppressed. However, it is not clear how PKP2 is functioning to downregulate these genes, it is also not clear if downregulation of cell genes has any effect on observed changes in adipogenic genes or adipocyte phenotype. PKP2 signalling cascade leading to gene expression changes is not studied (except few phosphorylation), DNA synthesis or proliferation is not measured. Therefore, causative relationship between PKP2, regulated genes and adipocyte phenotype is not established. These questions should be addressed experimentally.*

We carried out quantitative proteomics and in-depth proteome-wide analysis to identify functional variations following the knock-down of PKP2 in human preadipocytes and adipocytes (**Figure 4a**). This yielded further insight into the shift from normal conditions in fat cells enduring the loss of this scaffold protein, and provided a *bona fide* basis to better address the specific relevance of PKP2 in fully differentiated and precursor fat cells (**Figure 4b-i**, and supplemental **Table S4** and **Figures S4a-f** and **S5a-d**). On the other hand, the causative relationship between defective PKP2 and alterations affecting cell cycle-related genes is now better endorsed by results assessed in fat samples collected in a heterozygous *Pkp2*^{+/-} mouse model (**Figure 5a-b**) and adipose mesenchymal stromal cells of PKP2-dependent arrhythmogenic cardiomyopathy (ACM) patients (**Figure 5c-f**). We also provide the impact in cell growth (**Figure 1i**) and changes affecting oxidant production (related to the loss of nuclear envelope integrity and DNA

damage in PKP2-defective cardiac cells ⁷) in our cell models with altered PKP2 levels (**Figure 6g**). Finally, we examined the hypothesis that sets nuclear Protein Kinase C α (PKC α), a scaffold regulator linking signal transduction pathways and cell-cycle machinery ^{10,15} directly affected by defective PKP2 ¹¹, at the crossroad of inflammatory events affecting PKP2 signalling and the expression of cell cycle-related genes/proteins of apparent relevance to adipocyte function. As the Referee can see below and in the revised version of our manuscript, such experiments confirmed significant alterations in adipocyte biology driven by impaired PKP2 and potentially related to the subcellular location of PKC α (**Figure 6i** and **Figures S6e-f**). We believe that our efforts to set up multiple representative *in vitro* models for the study of this novel factor in the obesity arena, including the inclusion of results in different cell systems and exhaustive analysis of transversal/longitudinal human patient cohorts, represent a strength, reinforcing the causal relationship between our gene candidate and the activation of pathological mechanisms in obese/inflamed adipocytes, thus narrowing the relevance of this molecule in the field.

“Initially, we observed impaired growth prior to confluence (**Figure 1i**), recapitulating multiple lines of evidence linking PKP2 to cell proliferation ²⁵⁻²⁷.”

“We also evaluated changes in oxidant production, related to the loss of nuclear envelope integrity and DNA damage in PKP2-defective cardiac cells ⁶⁴, by assessing H₂O₂ in the media, and conducted immunofluorescence staining of the cell cycle driver Cyclin D1 (CCND1) ^{65,66} and Protein Kinase C α (PKC α), a scaffold regulator linking signal transduction pathways and cell-cycle machinery ⁶⁷ that is directly influenced by the lack of PKP2 ¹.”

“We further confirmed the increased presence of PAPPA at the protein level (**Figure 6f**), also accompanied by the high release of H₂O₂ into the culture media of PKP2-deficient adipocytes (**Figure 6g**), and found Cyclin D1 immunostaining to be significantly compromised (**Figure 6h**), while the appearance of nuclear PKC α (**Figure S6e-f**) was higher in adipocytes that had been challenged with synthetic oligonucleotides against PKP2, as well as in those under the influence of MCM (**Figure 6i**).”

2) *Second major point is discrepancy in adipogenic regulation of PKP2 and the effect of its knockdown. The upregulation of protein expression during adipogenesis is usually connected with the role of this protein in the function of mature fat cell. However, downregulation of PKP2 enhances formation of functional adipocyte. How this comes together? What is the mechanism for that? The effect on cell cycle and senescence genes does not explain this phenomenon and the authors don't make an attempt to explain it. This is definitely important for understanding PKP2 function in WAT.*

We thank the Reviewer for the opportunity to explain these counter-intuitive results. During the last few decades, investigators have embarked on a systematic endeavour to assess the biology of fat precursor cells and define the nature of lipid-containing adipocytes. In within, many of us struggle to comprehend the transcriptional events directing preadipocyte differentiation (adipogenesis), which is regulated by an intricate

network of transcription factors coordinating the expression of hundreds of proteins, defining the adipocyte phenotype¹⁹. Indeed, committed preadipocytes arise from multipotent mesenchymal progenitor cells found in adipose tissues, and show expression patterns and molecular features very distant from those in which terminally differentiated fat cells rely to perform their activity. As a matter of fact, sequential activation and altered expression of over 2,500 genes engage the entire terminal differentiation process^{20,21}. In this context, evidence compiled during our research points out two scenarios: first, preadipocytes with low PKP2 signal (e.g., sh/si-PKP2-treated and SC preadipocytes) show molecular insights of an emerging adipogenic phenotype, and are more prone to the adipogenic conversion than those with high amounts of PKP2 (OM preadipocytes). This observation aligns with a number of publications depicting the adipogenic component of uncommon cardiomyopathies diagnosed in patients with mutations affecting *PKP2*^{2,12}. Accordingly, the molecular basis of these diseases relies on cardiac progenitor cells that go awry and differentiate into adipocyte (phenotype-like) cells. Apparently, the pathogenesis of adipocytes replacing cardiac myocytes involves the partial suppression of canonical Wnt signalling^{12,22}, which stands for a strong deterrent of adipogenesis²³⁻²⁵. On the other hand, PKP2 is a key component of desmosomes necessary for maintaining tight junction-associated transmembrane communication, cell adhesion and interaction between neighbour cells, which are recognised requirements for adipogenesis²⁶⁻²⁸. In this context, increased PKP2 during adipogenesis should guarantee junction dynamics and fulfil the necessity of cell-to-cell contact, allowing adipocyte differentiation, commitment and survival in the complex landscape of the adipose tissue microenvironment. Therefore, a balance between the positive and negative regulation of PKP2 might be essential to have a controlled effect on adipogenesis as well as in adipocyte function, as diagrammatically represented now in **Figure 7**. Our results indicate that this balance is disrupted in obese subjects, thus compromising adipocyte wellbeing and ultimate viability. This leads to the second scenario, in which inflammatory events occurring in adipose tissue may drive defective PKP2 renewal, impaired desmosomal plaque assembly, hampered cell-to-cell contact, altered gene expression patterns, compromised cell function, and ensuing adipocyte senescence. Finally, it should be noted that there are some other examples of proteins upregulated during adipogenesis (and thus bounded to the phenotype of mature fat cells) whose downregulation enhances formation of functional adipocytes from precursor cells. For instance, the canonical Wnt/ β -catenin pathway, a well-recognised mitogen-driven cell cycle regulator and potent inhibitor of adipogenesis, plays in mature fat cells important roles in maintaining the adipocyte phenotype^{30,31}. Also, while PKC α levels are transiently elevated during the differentiation of 3T3-L1 and 3T3-F442A adipocytes, both the knockdown and chemical inhibition of this isoform (amongst others) can facilitate adipogenesis, suggesting the negative impact of this kinase in the development of new adipocytes (reviewed in²⁹). Our compiled results show consistent variations affecting the appearance of nuclear PKC α in adipocyte cultures in which PKP2 levels were modulated. These observations are consistent with data from different models whereby the interaction between PKP2 and PKC isoenzymes, either directly or in conjunction with other scaffolding proteins, has been reported^{11,32}. In these

precedents, the notion that intracellular translocation of activated PKC α may regulate molecular events linking signal transduction pathways to cell-cycle machinery is also sustained by compelling evidence readily available in the literature ^{33,34}. Thus, the underlying mechanism in adipocyte-specific PKP2 deficiency may comprise the translocation of nuclear PKC α , and supports the hypothesis that PKP2 prevents undesirable molecular patterns in adipocytes by locally tying PKC α to the cytoplasm. By doing so, PKP2 may avoid changes in the distribution of active PKC α and the phosphorylation of PKC nuclear substrates related to the synthesis of proteins involved in cytoskeletal dynamics, junction assembly, cell cycle, and adipocyte differentiation and maintenance. See now in the **Discussion** section of this revised manuscript:

“In these cell models, adipocytes presented attributes related to cellular senescence, including decreased amounts of the cell cycle driver cyclin D1, enhanced β -galactosidase activity and the subsequent release of H₂O₂, as well as changes in gene expression and protein biomarkers suggestive of impaired cell cycle, altered abundance of junctional proteins, and activated senescence, while the exogenous plasmid-based expression of human PKP2 seems to be sufficient to mitigate the chain of events leading to adipocyte failure. At the crossroad of these events able to further the molecular pathophysiology of obesity, we found consistent variations affecting the appearance of nuclear Protein Kinase C alpha (PKC α) in adipocyte cultures in which PKP2 levels were modulated. These results are consistent with data from different models whereby the interaction between PKP2 and PKC isoenzymes, either directly or in conjunction with other scaffolding proteins, was reported ^{1,86}. It is well recognised that calcium-activated, phospholipid-dependent PKC activity is involved in cellular functions ranging from growth control to intercellular adhesion, activation of cellular function, differentiation, and cell death ^{87,88}. The notion that intracellular translocation of activated PKC α regulates molecular events linking signal transduction pathways to cell-cycle machinery is also supported by compelling evidence readily available in the literature ^{89,90}. The underlying mechanism in adipocyte-specific PKP2 deficiency potentially comprises the translocation of nuclear PKC α , and supports the hypothesis that PKP2 prevents aberrant molecular patterns in adipocytes by locally tying PKC α to the cytoplasm, thus preventing changes in the distribution of active PKC α and the phosphorylation of PKC nuclear substrates related to the synthesis of proteins involved in cytoskeletal dynamics, junction assembly, cell cycle, and adipocyte differentiation and maintenance.”

3) *The authors show that PKP2 is equally expressed in SVF and adipocytes in scWAT and even higher in SVF in vsWAT. Having in mind that adipocytes constitute only around 30% of adipose tissue cells, all measurements of PKP2 expression in WAT don't tell much about PKP2 function in mature adipocytes. Therefore, it is unclear in which cell type PKP2 is regulated during obesity and how that relates to adipocyte phenotype. In this context, the results obtained from clinical data are not conclusive.*

We understand the scepticism of the Referee with regard to the significance of changes affecting PKP2 gene expression in bulk adipose tissue. Indeed, we must be very cautious when interpreting whole-tissue datasets, as major variations could arise from changes

in cell populations and/or molecular events affecting specific cell types. Certainly, results provided for SC PKP2 seem difficult to reconcile with the data acquired in OM adipose tissue. Yet, an explanation is plausible. Regarding the measurements of *PKP2* in isolated SVC and adipocytes, it should be noted that our own results and those of Vijay *et al.*³⁵ were obtained in morbid obese patients. In these subjects, adipocyte-specific *PKP2* is likely to be compromised, thus the lack of differences between SC/OM-derived adipocytes and SC SVC. On the other hand, high expression of *PKP2* in OM SVC mostly acknowledges the abundance of mesothelial cells within OM adipose tissue, while being almost absent in SC fat^{36,37}. The paradigm is nicely endorsed by Norreen-Thorsen *et al.* in reference³⁸. Such datasets served to illustrate in our revised supplemental **Figure S2e** the apparent overrepresentation of *PKP2* in SC adipocytes, while in OM adipose tissue this armadillo-repeat protein is prevalent in mesothelial cells. However, only with the analysis of extended population-based fat cell samples this relationship can be validated further. Unfortunately, isolated fat cells from extended human cohorts are not available at the moment, which compromises our ability to address the question as we would have liked to do. However, we are sensitive to the suggestion made by the Referee and this limitation has been noted in the **Results** section of our revised article (see below).

“Consistently, measures of *PKP2* gene expression taken in *ex vivo* isolated mature adipocytes (MA) and the stromal vascular cell (SVC) fraction of morbid obese patients were suggestive of different cell populations growing within OM (but not SC) SVC, as shown by our RT-PCR results (**Figure 2i**) and the single-cell RNA sequencing of Vijay *et al.*³⁵ (**Figure S2d**). This matches the enrichment of *PKP2* mRNA in SC adipocytes, while being more likely associated with the expression of biomarkers of mesothelial cells within OM fat (**Figure S2e**)³⁶. However, because these observations in bulk adipose tissue and adipose-derived cell samples were not population-based, we cannot exclude that the apparent relationship with obesity (results in SC) and other clinical characteristics (OM and SC) relies on the abundance of specific fat cell subgroups regulated during obesity. Nevertheless, our experiments *in vitro*, together with the systematic scrutiny of multiple human datasets and observations made in obese subjects following weight loss, support to a large extent the connexion between low *PKP2* in SC adipose tissue/ adipocytes and the burden of obesity/ inflammation. The exact mechanism whereby sex, age, and/or the metabolic status may influence the expression of *PKP2* in the even more complex cellularity of OM adipose tissue is a key question we seek to answer in our future studies.”

Some minor points:

- Figure 1. How many women were examined for the data shown in 1c. Preadipocytes from obese donors are known to differentiate worse than preadipocytes obtained from lean individuals. Data shown in 1c can simply be connected with lower differentiation capacity of omental preadipocytes. Please show other adipogenesis regulated genes, such as HSL, PLIN, ADIPOQ.

We thank the Referee for his/her kind suggestion. We have added plots depicting expression dynamics for well-recognised adipogenesis regulated genes (i.e., *ADIPOQ*, *PLIN*, and *FASN*). In **Figure 1c**, we show expression patterns for differentiating SC preadipocytes obtained from one lean (BMI<25 kg/m²) and one obese (BMI>30 kg/m²) women of approximately same age (35-40 years). We now specify it in the text. We are aware of the point made by the Referee concerning the important caveat that may represent inter-individual differences between fat precursor cells from different donors, and we cannot rule out the potential impact of other confounders, besides sex, age and weight. Only with the analysis of extended population-based cell cultures, these expression dynamics for *PKP2* in differentiating human adipocyte precursor cells can be validated further. Notwithstanding this, through these pioneering assays in primary adipocyte progenitors, we wanted to test whether *PKP2* was related to the adipogenic program, and how. Thereby, data shown in **Figure 1b** and **1c** indicate the unprecedented connection of *PKP2* with canonical adipogenesis, and thus, attenuated differentiation capacity of obese and omental preadipocytes further substantiated this association.

- *Please, show the kinetics of knockdown and overexpression efficiency of PKP2 by western (transient vs sustained).*

The kinetics of *PKP2* knockdown and overexpression efficiency at protein levels are shown in **Figures 3b, 6i** and **6h**.

- *English language should be checked and edited where needed.*

English language has been edited along the manuscript, and a few stylistic changes have been made to this revised version.

Reviewer #3 (Remarks to the Author):

In this research article, the Authors aimed at dissecting the role of Plakophilin-2 (PKP2) during adipogenesis in pre-adipocytes (PA), within the context of depot specific locations, including subcutaneous (SC) or omental (OM) PA. To this end, the Authors performed analyses of depot-related PKP2 gene expression in PA obtained from non-obese (NO), as well as lean or obese women, clustered on BMI bases, monitoring the PKP2 patterning in the resulting adipocytes. They investigated PKP2 expression during adipogenesis within the context of an in vitro induced pro-inflammatory microenvironment, as well as in 3T3-L1 fibroblasts during their commitment to an adipogenic lineage. The Authors investigated the consequences elicited by diminished (i.e. lentiviral induced) PKP2 expression in 3T3-L1 preadipocytes on their differentiation into lipid-containing fat cells. The pattern of PKP2 was followed in SC and OM adipocytes from obese and lean subjects, even assessing the consequences of BMI reduction owing to significant weight loss after bariatric surgery, and consequent reduction in caloric intake. The study was corroborated by wide ranging transcriptional profiling to identify the putative up or downregulated transcripts under the different experimental conditions. In summary, the Authors found that:

- PKP2 expression peaked in terminally differentiated adipocytes*
- A downregulation of PKP2 occurred in adipocytes under a pro-inflammatory microenvironment.*
- PKP2 had a significant role in maintaining a gene expression program required to sustain cell cycle.*
- PKP2 was downregulated in SC adipose cells from obese subjects, being normalized upon mild-to-intense weight loss.*
- Impaired PKP2 levels in human adipocytes (elicited by a pro-inflammatory context or artificial gene downregulation) inhibited cell cycle dynamics ensuing in premature senescence, and increased intracellular lipid accumulation.*
- Restoring PKP2 in inflamed adipocytes led to cell cycle re-activation, counteracting senescence progression.*

On the whole, the Authors' claim that "their findings bound the expression of PKP2 in fat cells to the physiopathology of obesity, and uncover a previously unknown defect in cell cycle and activated adipocyte senescence due to impaired PKP2" appears to be substantiated by the provided experimental evidence. The methodological approach is state-of-the-art.

Besides this, I have noticed at page 6 (lines 209-210): "and thus inversely associated with BMI and the expression of leptin (LEP) (Figure 2e)". In Figure 2, this is depicted in panel f (Figure 2f).

We apologize for the lack of clarity in the previous version of this manuscript and we thank the Reviewer for his/her kindly help and suggestions. We have now taken extensive care in data labelling and description of results. See now along the revised manuscript.

On the same page 6 (lines 211-214): "In contrast with results in SC adipose tissue, measures of PKP2 in OM fat revealed no relationship with BMI (Figure 2f-g, in the text, while this is shown in panels e, g), "yet slightly decreased expressions in non-obese subjects with IGT" (Figure 2f, indeed shown in panel e).

We have amended the mistake, and hope that our new data and the revised description of available results make this section of the manuscript much clearer.

As a minor remark, in my humble opinion the Authors should avoid using sentences that overemphasize their own results, like “seminal findings”, “strikingly”,...

We have balanced the tone of our narrative along this revised version of the manuscript.

REFERENCES

1. Pilato, C. A. *et al.* Isolation and Characterization of Cardiac Mesenchymal Stromal Cells from Endomyocardial Bioptic Samples of Arrhythmogenic Cardiomyopathy Patients. *J. Vis. Exp.* **132**, 57263 (2018).
2. Sommariva, E. *et al.* Cardiac mesenchymal stromal cells are a source of adipocytes in arrhythmogenic cardiomyopathy. *Eur. Heart J.* **37**, 1835–1846 (2016).
3. Grossmann, K. S. *et al.* Requirement of plakophilin 2 for heart morphogenesis and cardiac junction formation. *J. Cell Biol.* **167**, 149–160 (2004).
4. Sommariva, E. *et al.* Oxidized LDL-dependent pathway as new pathogenic trigger in arrhythmogenic cardiomyopathy. *EMBO Mol. Med.* **13**, e14365 (2021).
5. Demichev, V., Messner, C. B., Vernardis, S. I., Lilley, K. S. & Ralser, M. DIA-NN: neural networks and interference correction enable deep proteome coverage in high throughput. *Nat. Methods* **17**, 41–44 (2020).
6. García-Marqués, F. *et al.* A Novel Systems-Biology Algorithm for the Analysis of Coordinated Protein Responses Using Quantitative Proteomics. *Mol. Cell. Proteomics* **15**, 1740–1760 (2016).
7. Pérez-Hernández, M. *et al.* Loss of Nuclear Envelope Integrity and Increased Oxidant Production Cause DNA Damage in Adult Hearts Deficient in PKP2: A Molecular Substrate of ARVC. *Circulation* **146**, 851–867 (2022).
8. Musgrove, E. A., Lee, C. S., Buckley, M. F. & Sutherland, R. L. Cyclin D1 induction in breast cancer cells shortens G1 and is sufficient for cells arrested in G1 to complete the cell cycle. *Proc. Natl. Acad. Sci. U. S. A.* **91**, 8022–8026 (1994).
9. Guardavaccaro, D. *et al.* Arrest of G(1)-S progression by the p53-inducible gene PC3 is Rb dependent and relies on the inhibition of cyclin D1 transcription. *Mol. Cell. Biol.* **20**, 1797–1815 (2000).
10. Marini, N. J. *et al.* A pathway in the yeast cell division cycle linking protein kinase C (Pkc1) to activation of Cdc28 at START. *EMBO J.* **15**, 3040–3052 (1996).
11. Bass-Zubek, A. E. *et al.* Plakophilin 2: A critical scaffold for PKC α that regulates intercellular junction assembly. *J. Cell Biol.* **181**, 605–613 (2008).
12. Chen, S. N. *et al.* The hippo pathway is activated and is a causal mechanism for adipogenesis in arrhythmogenic cardiomyopathy. *Circ. Res.* **114**, 454–468 (2014).
13. Cerrone, M. *et al.* Plakophilin-2 is required for transcription of genes that control calcium cycling and cardiac rhythm. *Nat. Commun.* **8**, 106 (2017).
14. Mertens, C. *et al.* Nuclear particles containing RNA polymerase III complexes associated with the junctional plaque protein plakophilin 2. *Proc. Natl. Acad. Sci. U. S. A.* **98**, 7795–7800 (2001).
15. Livneh, E. & Fishman, D. D. Linking protein kinase C to cell-cycle control. *Eur. J. Biochem.* **248**, 1–9 (1997).
16. Arimoto, K. *et al.* Plakophilin-2 promotes tumor development by enhancing ligand-dependent and -independent epidermal growth factor receptor dimerization and activation. *Mol. Cell. Biol.* **34**, 3843–3854 (2014).
17. Wu, Y., Liu, L., Shen, X., Liu, W. & Ma, R. Plakophilin-2 Promotes Lung Adenocarcinoma Development via Enhancing Focal Adhesion and Epithelial-Mesenchymal Transition. *Cancer Manag. Res.* **13**, 559–570 (2021).
18. Hao, X.-L. *et al.* Plakophilin-2 accelerates cell proliferation and migration through activating EGFR signaling in lung adenocarcinoma. *Pathol. Res. Pract.* **215**, 152438 (2019).
19. Cristancho, A. G. & Lazar, M. A. Forming functional fat: a growing understanding of adipocyte differentiation. *Nat. Rev. Mol. Cell Biol.* **12**, 722–734 (2011).

20. Farmer, S. R. Transcriptional control of adipocyte formation. *Cell Metab.* **4**, 263–273 (2006).
21. Rosen, E. D. & MacDougald, O. A. Adipocyte differentiation from the inside out. *Nat. Rev. Mol. Cell Biol.* **7**, 885–896 (2006).
22. Garcia-Gras, E. *et al.* Suppression of canonical Wnt/ β -catenin signaling by nuclear plakoglobin recapitulates phenotype of arrhythmogenic right ventricular cardiomyopathy. *J. Clin. Invest.* **116**, 2012–2021 (2006).
23. Christodoulides, C., Lagathu, C., Sethi, J. K. & Vidal-Puig, A. Adipogenesis and WNT signalling. *Trends Endocrinol. Metab.* **20**, 16–24 (2009).
24. Ross, S. E. *et al.* Inhibition of adipogenesis by Wnt signaling. *Science* **289**, 950–953 (2000).
25. de Winter, T. J. J. & Nusse, R. Running Against the Wnt: How Wnt/ β -Catenin Suppresses Adipogenesis. *Front. cell Dev. Biol.* **9**, 627429 (2021).
26. Chen, Q. *et al.* Fate decision of mesenchymal stem cells: adipocytes or osteoblasts? *Cell Death Differ.* **23**, 1128–1139 (2016).
27. Schiller, P. C., D'Ippolito, G., Brambilla, R., Roos, B. A. & Howard, G. A. Inhibition of gap-junctional communication induces the trans-differentiation of osteoblasts to an adipocytic phenotype in vitro. *J. Biol. Chem.* **276**, 14133–14138 (2001).
28. Murakami, K. *et al.* Antiobesity Action of ACAM by Modulating the Dynamics of Cell Adhesion and Actin Polymerization in Adipocytes. *Diabetes* **65**, 1255–1267 (2016).
29. Schmitz-Peiffer, C. The tail wagging the dog--regulation of lipid metabolism by protein kinase C. *FEBS J.* **280**, 5371–5383 (2013).
30. Bagchi, D. P. *et al.* Wnt/ β -catenin signaling regulates adipose tissue lipogenesis and adipocyte-specific loss is rigorously defended by neighboring stromal-vascular cells. *Mol. Metab.* **42**, 101078 (2020).
31. Chen, M. *et al.* CTNNB1/ β -catenin dysfunction contributes to adiposity by regulating the cross-talk of mature adipocytes and preadipocytes. *Sci. Adv.* **6**, eaax9605 (2020).
32. Nagler, S. *et al.* Plakophilin 2 regulates intestinal barrier function by modulating protein kinase C activity in vitro. *Tissue barriers* 2138061 (2022).
33. Chalkiadaki, G. *et al.* Low molecular weight heparin inhibits melanoma cell adhesion and migration through a PKCa/JNK signaling pathway inducing actin cytoskeleton changes. *Cancer Lett.* **312**, 235–244 (2011).
34. Martelli, A. M., Sang, N., Borgatti, P., Capitani, S. & Neri, L. M. Multiple biological responses activated by nuclear protein kinase C. *J. Cell. Biochem.* **74**, 499–521 (1999).
35. Vijay, J. *et al.* Single-cell analysis of human adipose tissue identifies depot and disease specific cell types. *Nat. Metab.* **2**, 97–109 (2020).
36. Chau, Y.-Y. *et al.* Visceral and subcutaneous fat have different origins and evidence supports a mesothelial source. *Nat. Cell Biol.* **16**, 367–375 (2014).
37. Emont, M. P. *et al.* A single-cell atlas of human and mouse white adipose tissue. *Nature* **603**, 926–933 (2022).
38. Norreen-Thorsen, M. *et al.* A human adipose tissue cell-type transcriptome atlas. *Cell Rep.* **40**, 111046 (2022).